# Proper Orthogonal Decomposition for Scalable Training of Graph Neural Networks

**Abhishek A[†], Manohar Kaul, Mohit Meena, Mahesh Chandran**
*Fujitsu Research of India, Bangalore, India*
*{abhishek.a, kaul.manohar, mohitkumar.meena, mahesh.chandran}@fujitsu.com*

Reviewed on OpenReview: *https://openreview.net/forum?id=LeL6whBoWE*

## Abstract

As large-scale graphs become ubiquitous in real-world applications, there is growing concern about the memory and time requirement to train a graph neural network (GNN) model for such datasets. Storing the entire adjacency and node embedding matrices in memory is infeasible in such a scenario. Standard sampling-based methods for addressing the memory constraint suffer from the dependence of the number of mini-batches on the graph size. Existing sketch-based methods and graph compression techniques operate at higher sketch ratios, with the graph compression techniques showing poor generalization, implying that different GNNs trained on the same synthetic graph have performance gaps. Sketch-based methods necessitate online learning of sketches, further increasing the complexity. In this paper, we propose a new sketch-based algorithm, PGNN, employing the Proper Orthogonal Decomposition (POD) method to craft update rules to train GNNs, improving the memory requirement and training time without the complication of updating the sketches during training. Experiments on standard graph datasets show that PGNN can reach much lower sketch ratios without compromising the performance. We demonstrate that the POD projection matrix is provably optimal, minimizing an upper bound on the projection error induced by the linearized GNN (SGC) update rule. Empirical findings validate our approach, demonstrating superior performance at reduced sketch ratios and adaptability across various GNN architectures.

## 1 Introduction

Graph Neural Networks (GNNs) have proven to be powerful tools for graph learning across various domains, excelling in tasks such as classification (Kipf & Welling, 2017), clustering (Bianchi et al., 2020), recommendation systems (Wu et al., 2022), and social network analysis (Fan et al., 2019). Their strength lies in their ability to extract meaningful insights from local neighbourhoods within graphs, thus creating effective representations of target nodes. However, the dependence of GNNs on graph topology introduces significant challenges when scaling to larger graphs or deeper models while maintaining computational and memory efficiency. Traditional full-batch training methods necessitate storing the Laplacian matrix of the entire graph, resulting in a memory complexity of $O(m + ndL + d^2L)$ for an $n$-node, $m$-edge graph, where node features have dimension $d$ in an $L$-layer *graph convolutional network* (GCN). This linear dependency on both $n$ and $m$, combined with the limited memory capacity of GPUs, restricts the scalability of training on large graphs (especially large dense graphs with $m$ being of the order of $O(n^2)$ in the worst case). To address these memory constraints, research in this domain has broadly proposed two main approaches: sampling-based strategies (Hamilton et al., 2018; Chen et al., 2018a;b; Chiang et al., 2019; Zeng et al., 2020) and historical embedding techniques (Fey et al., 2021; Ding et al., 2021). Although these methods improve memory efficiency, the computational complexity still increases linearly with $n$ and $m$.

---

[†]Corresponding author.

In the context of matrix approximation, a *sketch* of an arbitrary matrix $A \in \mathbb{R}^{n \times d}$ is defined as a reduced matrix $B \in \mathbb{R}^{c_0 \times d}$, where $c_0$ denotes the *sketch dimension*, and $B$ providing a good approximation to $A$ even as $c_0 \ll n$(Ghashami et al., 2015). The amount of compression achieved by sketching is best described by the *sketch ratio $r = c_0/n$. Proper orthogonal decomposition* (POD), as described in the article Rathinam & Petzold (2003), also known as the Karhunen–Loéve decomposition or principal component analysis, provides an orthonormal basis representing the given data in an optimal least squares sense. To achieve sublinear training time complexity with respect to $n$, Ding *et al.* (Ding et al., 2022) propose a sketch-based algorithm named *sketch-GNN*, that trains the GNN on top of a few compact sketches of both the *convolution* and *node feature* matrices. The authors propose an end-to-end training protocol in the sketch space by approximating the non-linear activation function using *polynomial tensor-sketch* (PTS) theory (Pham & Pagh, 2013a).

As observed by the authors, the approximation of the non-linear activation limits the expressiveness, which constrains the depth of GNNs that can be trained due to error accumulation. Working with dense matrices, even in the reduced sketch space, imposes a significant computational burden. Despite showing promise, the sketch-GNN algorithm has following limitations: (i) it requires *higher sketch ratio* which results in a lower compression of the original graph and (ii) it *requires frequent updates* of the sketches during the training which triggers re-computation of all the sketches involved. It is worthwhile to mention another technique which guarantees efficient training of GNNs using random spanning trees (Bonchi et al., 2024) and leverages the concept of effective resistance to enhance node classification tasks. This method enhances GNN efficiency by creating path graphs from random spanning trees to maintain essential graph features while minimizing complexity for faster training. This approach is currently constrained to only the GCN architecture.

The primary motivation for using POD to address limitation (i) in sketch-GNN was driven by the parallel between dynamical systems and message passing methods in GNN Choi et al. (2023), coupled with the fact that effective number of eigenmodes required for POD in dynamical system decreases on increasing system size, resulting in a much lower sketch ratio. For limitation (ii), we theoretically establish bounds that constrain the deviations of node representations when using our proposed method. Additionally, we prove the optimality of the POD update rule for a projection subproblem induced by the linearized GNN update rule. This indicates that the best low-rank matrix for the update rule can be predetermined, which completely eliminates the need for online learning via frequent updates of sketches. To this end, our paper presents **PGNN**, a novel sketch-based method for GNNs. This method diverges fundamentally from prior approaches that emphasize sketching weights or gradients (see Liu et al. (2022); Chen et al. (2015); Kasiviswanathan et al. (2018); Lin et al. (2019); Spring et al. (2019)). Drawing inspiration from the update rules of linearized GNNs (SGC) (Wu et al., 2019), we customize the message passing process to function within the linear subspace formed by the columns of the augmented input node feature matrix. Experimental results, as presented in section 5, demonstrate that the sketch ratio necessary for achieving optimal performance decreases as the graph size increases. Despite its theoretical limitations beyond linearized GNN, the PGNN framework efficiently performs node classification in POD-derived linear subspaces, providing insights into GNN operational subspaces (Lee et al., 2023). Our contributions can be summarized as follows:

1. In section 4, we introduce specialized update rules designed to enhance the training efficiency of GNNs by operating within a reduced subspace. Utilizing the POD method, we sketch the input node feature and convolution matrices into their lower-dimensional approximations, thereby streamlining the computational process.

2. In Theorem 1, we establish that the POD projection matrix minimizes an upper bound on the projection error for the linearized GNN update rule. Furthermore, in Theorem 2, we present bounds that quantify the deviation of node representations when using the PGNN framework.

3. The versatility of PGNN is evaluated across different GNN architectures, including GCN (Kipf & Welling, 2017), SGC (Wu et al., 2019), GraphSAGE (Hamilton et al., 2018), and GAT (Veličković et al., 2018), with results detailed in section 5. Through extensive experimentation, as demonstrated in section 5, we find that PGNN is able to provide comparable performance at much lower sketch-ratio compared to previous methods. For instance, on the Reddit dataset, the state-of-the-art sketch-GNN framework achieves its highest accuracy at a sketch ratio of 0.3, while we achieve better and comparable accuracy for various architectures for a much lower sketch ratio of 0.05. This

reduction in sketch ratio directly translates to accelerated runtime and a reduced memory footprint, underscoring our method's efficiency without compromising accuracy.

## 2 Related work

The scalability of GNNs has been predominantly addressed through mini-batching strategies, which, despite mitigating memory bottlenecks, often fail to reduce epoch training time. Recent work in graph compression, such as Graph Coarsening (Loukas, 2018) and dataset condensation (Zhao et al., 2021), aims for sublinear training times by condensing the graph, thus reducing node and edge counts (Huang et al., 2021; Jin et al., 2022). These methods, however, face significant challenges: the preprocessing overheads often exceed $O(n)$, reducing practical benefits, and the efficacy of the trained model varies with the GNN architecture used (Jin et al., 2022; Ding et al., 2022). Scalable GNN approaches fall into several categories: (A) full-graph training, which is memory and time-intensive; (B) sampling-based methods like GraphSAGE (Hamilton et al., 2018), FastGCN (Chen et al., 2018b), and GraphSAINT (Zeng et al., 2020), which employ various sampling strategies to reduce computational load; (C) historical-embedding methods, such as GNNAutoScale (Fey et al., 2021) and VQ-GNN (Ding et al., 2021), which store embeddings but incur high memory costs; (D) linearized GNNs (Bojchevski et al., 2020; Wu et al., 2019; Frasca et al., 2020), which offer computational efficiency at the risk of oversimplification; (E) methods using random spanning trees (Bonchi et al., 2024), which reduce computational load by transforming graphs into sparse path graphs; and (F) sketch-based methods like Sketch-GNN (Ding et al., 2022), which approximate non-linear activations but struggle with error accumulation and high computational demands. Each approach presents trade-offs in terms of computational complexity and model expressiveness, addressing different constraints in GNN applications.

## 3 Preliminaries

**Basic Notations.**

Let $G = (V, E)$ denote a graph where $V = [n] := \{1, \ldots, n\}$ is the set of $n$ vertices and $E \subset V \times V$ is the set of $m$ edges. $y$ represents the labels of the nodes. Additionally, the input node feature matrix associated with $G$ is denoted by $X^{(0)} \in \mathbb{R}^{n \times d}$, where $d$ is the number of features. Let $\tilde{X}^{(0)} \in \mathbb{R}^{n \times c_0}$ and $\bar{x}$ denote the *augmented feature matrix* and its *mean vector*, respectively. $C \in \mathbb{R}^{n \times n}$ denotes the convolution matrix of graph $G$ and $C(i, j)$ denotes its $(i, j)$-th entry. We represent the $k^{th}$ order element-wise power of $C$ as $C^{\odot k}$. Additionally, $C(i, :)$ denotes the $i^{th}$ row and $C(:, j)$ denotes the $j^{th}$ column. Considering a GNN, $X^{(l)} \in \mathbb{R}^{n \times d_l}$ denotes the node representations of layer $l$, where $d_l$ represents the number of neurons at layer $l$. $\|\cdot\|$ denotes the $\ell_2$ norm unless stated otherwise. $\sigma(\cdot)$ is the non-linear activation and $\Theta^{(l,q)}$ is the learnable weight matrix at layer $l$ for filter $q$. $R^{(1)}, R^{(2)}, \ldots, R^{(k)}$ denote the $k$ count-sketch matrices with dimension $\mathbb{R}^{c_k \times n}$, where $k$ (a hyper-parameter) denotes the number of sketches and $c_k$ is the fixed sketch dimension associated with all of them. $\beta$ denotes the upper bound on the number of elements in the set for unsketching. The POD projection matrix which is the matrix of the linear projection expressed in the original coordinate system in $\mathbb{R}^n$ is given by $P = \rho^T \rho \in \mathbb{R}^{n \times n}$. We refer to the submatrix $\rho$ as the *factor* of $P$. We represent a matrix comprising of $b$ $n$-dimensional column vectors $y \in \mathbb{R}^n$ as $[y]_{n \times b}$. $\rho_1 \in \mathbb{R}^{c_0 \times c_0}$ denotes the learnable matrix incorporated in the PGNN update rule. $\beta_2$ denotes the number of nodes used for unsketching to update $\rho_1$.

### 3.1 Count Sketch.

Matrix multiplication is crucial in machine learning and scientific computation, with efficient techniques developed in works like (Paszke et al., 2017; Guennebaud et al., 2010), and Abadi et al. (2016). Count sketch, a potent dimensionality reduction technique introduced in Charikar et al. (2002) and Weinberger et al. (2010), projects an $n$-dimensional vector $u$ into a $c_k$-dimensional space using a random hash function $h : [n] \to [c_k]$ and a binary Rademacher variable $s : [n] \to \{-1, 1\}$. The dimension reduction transformation $CS(u)_i = \sum_{h(j)=i} s(j)u_j = R(i, :)u$ involves a count sketch matrix $R \in \mathbb{R}^{c_k \times n}$. The work in Ding et al. (2022) provide theoretical guarantees on the approximation quality of CountSketch for Graph Neural Networks; we restate the relevant result in Appendix A (Lemma 2).

### 3.2 Tensor Sketch.

Tensor sketch, introduced as a generalization of the count sketch Charikar et al. (2002), is a dimensionality reduction technique frequently employed in machine learning for large datasets Pham & Pagh (2013b). Pham and Pag Pham & Pagh (2013a) proposed an efficient way to compute tensor-sketch using FFT and inverse FFT operation given by,

$$\text{TS}_k(A) = \text{FFT}^{-1} \bigodot_{p=1}^{k} \text{FFT}\left(\text{CS}^{(p)}(A)\right).$$

In Ding et al. (2022), Tensor-sketch is applied to approximate the element wise $k$-th power of a matrix product:

$$(AB)^{\odot k} \approx \text{TS}_k(A)\,\text{TS}_k(B^T)^T,$$

where $A \in \mathbb{R}^{n \times n}$, $B \in \mathbb{R}^{n \times d}$, and $\text{TS}_k(A) \in \mathbb{R}^{n \times c_k}$, $\text{TS}_k(B^T) \in \mathbb{R}^{d \times c_k}$ with $c_k < n$.

### 3.3 Locality Sensitive Hashing.

Locality Sensitive Hashing (LSH) exploits hash functions, denoted as $H : \mathbb{R}^d \rightarrow [c_k]$, to map closely positioned vectors into the same bucket with high probability. SimHash, an instance of LSH, uses a random matrix $P \in \mathbb{R}^{c_k/2 \times d}$ to define a hash function $H(u) = \arg\max\left([Pu\,||-Pu]\right)$ (Charikar et al., 2002). This method is efficient for large vector batches (Andoni et al., 2015).

### 3.4 Proper Orthogonal Decomposition.

Given the input node feature matrix $X^{(0)} = [x_1, x_2, \dots, x_d]$, where $x_i \in \mathbb{R}^n$. Then the best approximating affine subspace representing these data points and passing through the mean ($\bar{x} = \frac{1}{d}\sum_{i=1}^{d} x_i$) is given by the leading eigenvectors of the *centred covariance matrix* (see (Rathinam & Petzold, 2003) for a detailed explanation)

$$\bar{R} = \frac{1}{d-1}\sum_{i=1}^{d}(x_i - \bar{x})(x_i - \bar{x})^T.$$

The factor $\rho \in \mathbb{R}^{c_0 \times n}$ of projection $P$ is given by the leading eigenvectors of $\bar{R}$, where $c_0 \ll n$. The sketch $Z^{(0)} = \rho(X^{(0)} - [\bar{x}]_{n \times d}) \in \mathbb{R}^{c \times d}$ of input node feature matrix $X^{(0)}$ represents the sketch of $X^{(0)}$ in the affine subspace. Details of the POD technique can be found in (Rathinam & Petzold, 2003; Holmes et al., 1996; Lall et al., 1999; Moore, 1981).

### 3.5 Unified Framework of GNNs.

For a GNN, message passing between layers can happen differently, like that of spatial convolution (GCN)(Kipf & Welling, 2017), self-attention (GAT)(Veličković et al., 2018), and Weisfeiler-Lehman (WL) alignment, see Xu et al. (2019). According to Balcilar et al. (2021) the general rule for message passing is given by,

$$X^{(l+1)} = \sigma\left(\sum_q C^{(l,q)} X^{(l)} \Theta^{(l,q)}\right), \tag{1}$$

where $C^{(l,q)} \in \mathbb{R}^{n \times n}$ is the $q$-th convolution support at layer $l$ that defines how the node features are propagated to the neighbouring nodes, $X^{(l)}$ is the node representations at layer $l$, and $\Theta^{(l,q)}$ are the trainable weights. The input node feature matrix is given by $X^{(0)} \in \mathbb{R}^{n \times d}$.

As shown in Ding et al. (2021), the gradient involved in the back-propagation rule for GNNs, for the loss function $\ell$, is given by the following:

$$\nabla_{X^{(l)}}\ell = \sum_q \left(C^{(l,q)}\right)^\top \left(\nabla_{X^{(l+1)}}\ell \odot M^{(l+1)}\right)\left(\Theta^{(l,q)}\right)^\top, \tag{2}$$

where $M^{(l+1)} = \sigma' \left( \sigma^{-1} \left( X^{(l+1)} \right) \right)$. This formulation embodies the essence of the message-passing paradigm. Here, $\sigma'$ and $\sigma^{-1}$ denote the derivative and the inverse of the activation function $\sigma$, respectively. The term $\nabla_{X^{(l+1)}} \ell \odot \sigma' \left( \sigma^{-1} \left( X^{(l+1)} \right) \right)$ represents the gradients propagated back through the non-linearity. In essence, this rule captures the flow of information and updates dynamics within GNNs during the backward pass.

## 4 POD sketch based method on GNNs

**Problem and Insights.** The runtime complexity of the update rules of GNNs on a complete graph is $O(n^2)$, and the memory complexity involved is $O(n + m)$. The POD sketch-based method for GNNs approximates the GNN's update rule and utilizes sketches of both the convolution matrix and the input node feature matrix for training. Initially, the input node feature matrix ($X^{(0)}$) and the convolution matrix ($C$) are of sizes $n \times d$ and $n \times n$, respectively. These matrices are then transformed into low-dimensional sketches of size $c_0 \times d$ and $c_0 \times c_0$, respectively. The sketch $Z^{(0)}$ of the input node feature matrix $X^{(0)}$ and the *convolution matrix sketch* ($S_C$) approximating the SGC architecture of GNN (PSGC) are given by:

$$Z^{(0)} = \rho(X^{(0)} - [\bar{x}]_{n \times d}), \quad S_C = \rho C \rho^T.$$

Figure 1 illustrates the overall PGNN framework. Recall that $\rho$ is the *factor* of $P$, representing the singular vectors of the augmented input node feature matrix $\tilde{X}^{(0)}$ after normalization (see Algorithm 2).

### 4.1 Approximate update rules with PGNN

Our primary goal is to approximate the forward propagation rule of the GNN:

$$X^{(l+1)} = \sigma \left( C X^{(l)} \Theta^{(l)} \right).$$

We project the node representations at layers $l$ and $l + 1$ onto the subspace spanned by the columns of the factor matrix $\rho$. This yields

$$Z^{(l+1)} = \rho \sigma \left( C \left( \rho^T Z^{(l)} + [\bar{x}]_{n \times d_l} \right) \Theta^{(l)} \right) - U.$$

The mean of the augmented input feature matrix $\tilde{X}^{(0)}$ (See Algorithm 2) is denoted by $\bar{x}$. We denote the bias induced by this projection as $U = [\rho \bar{x}]_{c_0 \times d_{l+1}}$. For ease of notation let the unsketched node representations at layer $l$ be $\hat{X}^{(l)} = \rho^T Z^{(l)} + [\bar{x}]_{n \times d_l}$. Employing an element-wise nonlinearity expressed as a power series (see Equation 4 in Ding et al. (2022)) leads to the following result:

$$Z^{(l+1)} = \rho \left( \sum_{k=1}^{q} c_k \left( C \hat{X}^{(l)} \Theta^{(l)} \right)^{\odot q} \right) - U.$$

Using Tensor-sketch (Section 3.2) to approximate the power series above leads to the following:

$$Z^{(l+1)} \approx \sum_{k=1}^{q} c_k S_C^k \left[ \mathrm{TS}_k \left( \hat{X}^{(l)} \Theta^{(l)} \right)^T \right]^T - U.$$

$c_k$ represents the learnable coefficients for combining different powers of the representation matrix. Sketch of the convolution matrix $S_C^k = \rho \, \mathrm{TS}_k(C)$.

$$Z^{(l+1)} = \sum_{k=1}^{q} c_k S_C^k \left( T S_k \left( \hat{X}^{(l)} \Theta^{(l)} \right)^T \right)^T - U$$

$$Z^{(l+1)} = \sum_{k=1}^{q} c_k S_C^k \left[ \mathrm{FFT}^{-1} \left( \bigodot_{p=1}^{k} \mathrm{FFT} \left( V_N^{(k)} \right) \right) \right]^T - U \tag{3}$$

$N = \hat{X}^{(l)}\Theta^{(l)}, V_N^{(k)} = \text{CS}^{(k)}(N^T) = N^T R^{(k)T}$, with $R^{(k)} \in \mathbb{R}^{n \times c_k}$ denoting the count-sketch matrix. The **advantage** of the PGNN update rule is that the objective is restricted to get optimal sketches of the matrix $\rho$, which will be fixed throughout training. In **Sketch-GNN**, the update rule involves *online learning of sketches* where the LSH and count-sketch hash tables corresponding to each layer are updated during training. The update rule consists of computing the count-sketch of matrix $\rho$. Count-sketch of matrix $\rho$,

$$\text{CS}^{(k)}(\rho) = \rho R^{(k)T} = \tilde{\rho}^{(k)}.$$

For the unsketching process, the count-sketch matrix $R^{(k)} \in \mathbb{R}^{c_k \times n}$ is transferred to GPU memory. Each column of $R^{(k)}$ contains a single nonzero entry—either $+1$ or $-1$ located at a random row. Storing the count-sketch matrix in memory is not an overhead because of its inherent sparse nature. To illustrate, for the ogbn-products dataset, a single count-sketch matrix consumes approximately 88 MB of memory for a count-sketch ratio of 0.1. The count-sketch ratio's dependence on the approximation's quality is addressed in Lemma 2. However, an additional storage cost of $O(c_0 c_k)$ is incurred to store the sketches $\tilde{\rho}$ and $R^{(k)}$. The intricacies of how message passing happens for the PGNN framework in various GNN architectures like SGC, GCN, GraphSAGE, and GAT are explained in Appendix B. Two challenges must be addressed for the approximate update rule proposed in Equation 3.

### 4.1.1 Challenges

**Challenge (1).** The PGNN method limits the update rule of the GNNs to the column space of $\rho$.
**Addressing challenge (1)** The PGNN update rule projects node representations onto the column space of $\rho$. To overcome this limitation, we introduce a learnable matrix $\rho_1 \in \mathbb{R}^{c_0 \times c_0}$ into the update rule (Equation 3), resulting in the following formulation:

$$Z^{(l+1)} = \sum_{k=1}^{q} c_k \, \rho_1 \, S_C^k \left[ \text{FFT}^{-1} \left( \bigodot_{p=1}^{k} \text{FFT} \left( V_{\tilde{N}}^{(p)} \right) \right) \right]^T - U. \tag{4}$$

$V_{\tilde{N}}^{(p)} = \left( \text{CS}^{(p)}(\tilde{N}^T) \right)$, $\tilde{N} = (\rho^T \rho_1 Z^{(l)} + [\bar{x}]_{n \times d_l})\Theta^{(l)}$. Equation 4 serves as the **update rule for PGNN**, approximating the original GNN update mechanism. The complexities associated with the PGNN update rule are explained in Appendix C. We design a loss function $\mathcal{L}$ to update the parameter $\rho_1$. This loss function incorporates the GNN loss $\ell$, which is evaluated on a subset $S_T$ of the training set. The subset $S_T$ consists of $\beta_2$ nodes, where $\beta_2$ is a hyperparameter chosen to be significantly smaller than the total number of training samples. The matrix $D = \rho^T(S_T, :)\rho_1 Z^{(0)} + [\bar{x}]_{\beta_2 \times d} - X^{(0)}(S_T, :)$,

$$\mathcal{L} = \underbrace{\alpha_0 \|D\|_F^2}_{\textbf{Term 1}} + \underbrace{\beta_0 \, \ell(S_T)}_{\textbf{Term 2}}. \tag{5}$$

$Z^{(L)}$ denotes the sketched node representations at the last layer and the unsketched node representations at the last layer $L$, $\hat{X}^{(L,k)}(S_T, :) = R^{(k)T}(S_T, :) \tilde{\rho}^{(k)T} \rho_1 Z^{(L)} + [\bar{x}]_{\beta_2 \times d_L}$. **Term 1** is used to control the deviations between the unsketched node representations at the input and the input node feature matrix, whereas **Term 2** gives weightage to the loss of the GNN $\ell$ while updating $\rho_1$. The *compute* needed to update $\rho_1$ using the loss function $\mathcal{L}$ is $O(\beta_2 c_k d_L + c_k c_0 d_L)$. $\alpha_0$ and $\beta_0$ are kept as constants. $\mathcal{L}$ is updated using gradient-descent on every epoch (See Algorithm 1).
**Challenge (2).** Avoiding $O(n)$ **in the loss evaluation.** **Unsketching** of node representations $Z^{(L)} \in \mathbb{R}^{c_0 \times d_l}$ at layer $L$ from the sketch dimension $c_0$ to $n$ and computing the losses for all nodes in node classification involves $O(c_k c_0 n)$ computations and $O(n)$ memory.

$$\hat{X}^{(L)} = \text{Mean} \left\{ R^{(k)T} \left( (\tilde{\rho}^{(k)})^T \rho_1 Z^{(L)} \right) \right\} + [\bar{x}]_{n \times d_l}. \tag{6}$$

Mean refers to the element-wise mean over tensors.
**LSH-based Loss Evaluation for Node Classification:** To avoid $O(n)$ complexity in loss evaluation, we employ a locality-sensitive hashing approach that selects nodes with poor predictions as described in Section 3.3 of Ding et al. (2022), outlined below:

1. **Construct LSH hash tables**: We build LSH hash tables $H : \mathbb{R}^d \to [c_k]$ (Section 3.3) to index the labeled training nodes across $C$ classes into $c_k$ hash buckets.

2. **Formation of subset** $B$: Using the LSH hash tables, we select nodes whose predicted class scores have small inner products with respect to their ground truth (one-hot encoded) labels based on the gradient signals of $M^{(L,k)} = \tilde{\rho}^{(k)^T} \rho_1 Z^{(L)}$.

$$B = \bigcup_{j=1}^{C} \left\{ \arg\max_j M_{:,j}^{(L,k)} \right\}$$

$$\hat{X}^{(L)}(B,:) = \text{Mean}\left\{ R^{(k)^T}(B,:) M^{(L,k)} \right\} + [\bar{x}]_{|B| \times C} \tag{7}$$

3. **Evaluate loss on selected subset**: We compute the classification loss only on the selected nodes:

$$\mathcal{L}_{\text{LSH}} = \frac{1}{|B|} \sum_{i \in B} \ell(\hat{y}_i, y_i) \tag{8}$$

$\ell$ is the loss function (e.g., cross-entropy), $\hat{y}_i$ represents the predicted class probabilities.

**Complexity advantage**: This approach avoids the $O(n)$ complexity of evaluating losses for all nodes by focusing computational resources on nodes that are most likely to contribute significant gradients, i.e., those with poor current predictions.

**Simhash projection matrix update.** The projection matrix $P$ for the SimHash function $H : \mathbb{R}^d \to [c_k]$ is updated using gradient descent with the triplet loss function (Equation 7 in Ding et al. (2022)), originally introduced in Chen et al. (2021).

$$\mathcal{L}_1(H, \mathcal{P}_+, \mathcal{P}_-) = \max\left\{ 0, \sum_{(u,v) \in \mathcal{P}_-} \cos(H(u), H(v)) - \sum_{(u,v) \in \mathcal{P}_+} \cos(H(u), H(v)) + \alpha \right\} \tag{9}$$

$$\mathcal{P}_+ = \left\{ (\hat{X}_{i,:}, \hat{X}_{j,:}) \mid i, j \in B, \langle \hat{X}_{i,:}, \hat{X}_{j,:} \rangle > t_+ \right\},$$

$$\mathcal{P}_- = \left\{ (\hat{X}_{i,:}, \hat{X}_{j,:}) \mid i, j \in B, \langle \hat{X}_{i,:}, \hat{X}_{j,:} \rangle < t_- \right\}$$

are the similar and dissimilar node-pairs in the subset $B$; $t_+ > t_-$ and $\alpha > 0$ are hyper-parameters. This triplet loss $\mathcal{L}_1(H, \mathcal{P}_+, \mathcal{P}_-)$ is used to update $P$ using gradient descent at every $T_{LSH}$ epoch. The complexities associated with the loss $\mathcal{L}_1$ are explained in Appendix C. In this section, we establish the theoretical foundation of our approach by presenting a key result on the optimality of the POD projection matrix in the context of a projection subproblem induced by linearized GNNs as shown in Theorem 1. Theorem 1 suggests that the POD method offers an optimal projection matrix for the linearized GNN update rule. Additionally, we analyze the error propagation in node representations within the PGNN framework across different layers, as formalized in Theorem 2.

**Theorem 1.** *Let $B = C^{(l)}\rho^T$, with $\Psi \in \mathbb{R}^{c_0 \times d}$ and $\Psi_1 \in \mathbb{R}^{n \times d}$ denoting arbitrary matrices. Consider the minimization problem*

$$\min_{Q \in \mathcal{P}} \|QB\Psi - B\Psi\|_F^2, \tag{10}$$

*where $\mathcal{P}$ denotes the set of rank $c_0$ orthogonal projection matrices. If column space of $B$ is a subset of column space of matrix $\rho^T$, there exists $\omega_i \in \mathbb{R}^{c_0}$ such that $PB\Psi(:,i) = F\omega_i$. $P = \rho^T \rho$ denotes the POD projection matrix, and $\rho^T$ the left singular eigenvectors associated with $F = \frac{\tilde{X}^{(0)} - [\bar{x}]_{n \times c_0}}{\sqrt{nc_0 - 1}}$. $\tilde{X}^{(0)}$ denotes the normalized augmented input node feature matrix, $\bar{x}$ denotes the mean along the columns of matrix $\tilde{X}^{(0)}$. Consequently, the objective function equation 10 satisfies the inequality*

$$\|QB\Psi - B\Psi\|_F^2 \leq \|QF - F\|_F^2 \sum_{i=1}^{c_0} \|\omega_i\|_2^2. \tag{11}$$

*The upper-bound of the objective function, given by equation 11 is minimized by the matrix $P$. (see proof in Appendix A.1)*

### 4.1.2   Error bound on the node representations at each layer $l$

**Theorem 2.** *Let $X^{(l)}$ and $\hat{X}^{(l)}$ represent the actual and approximate node representations for the PGNN method with the linearized GNN architecture at a layer $l$. Following the update rule $X^{(l+1)} = CX^{(l)}\Theta^{(l)}$, the normalized error $\epsilon^{(l+1)} = \dfrac{\left\|X^{(l+1)} - \hat{X}^{(l+1)}\right\|}{\left\|X^{(l+1)}\right\|}$ at layer $l+1$ caused by the PGNN method is given by,*

$$\epsilon^{(l+1)} \leq \frac{\|C - C_{eq}\|\,\|\Theta^{(l)}\|}{S} + \frac{\epsilon^{(l)}\,\|C_{eq}\|\,\|\Theta^{(l)}\|}{S} + \bar{T}$$

*where $\bar{T} = \left\|(I - P)[\bar{x}]_{n \times d_{(l+1)}}\right\|$ and the equivalent convolution matrix for the PGNN method $C_{eq} = PC, P = \rho^T \rho,\ (S = \dfrac{\left\|X^{(l+1)}\right\|}{\left\|X^{(l)}\right\|})$. (See proof in Appendix A).*

Theorem 2 indicates the quality of approximations made by the PGNN method depends on the equivalence of matrices $C$ and $C_{eq}$. The Theorem indicates that errors across layers accumulate with depth. We empirically study the propagation of error in deep GNNs in Appendix D.1 and observed that using the Jumping Knowledge framework (Xu et al., 2018) on PGNN compensates for accuracy loss and ensures faster convergence (Figure 8). The learned representations can be qualitatively assessed by visualizing the t-SNE plot of the features from the first layer of a pre-trained PGCN model, for example, shown for the Cora dataset in Figure 10. The visualization reveals distinct clusters in the 2D projected space. These clusters align with the seven labels of the dataset, demonstrating the model's ability to distinguish between the seven topic classes in Cora effectively. Appendix D.4 is dedicated to empirical validation, wherein a series of experiments are conducted to ascertain the congruence of the convolution matrices for the Cora dataset.

### 4.2   Algorithm

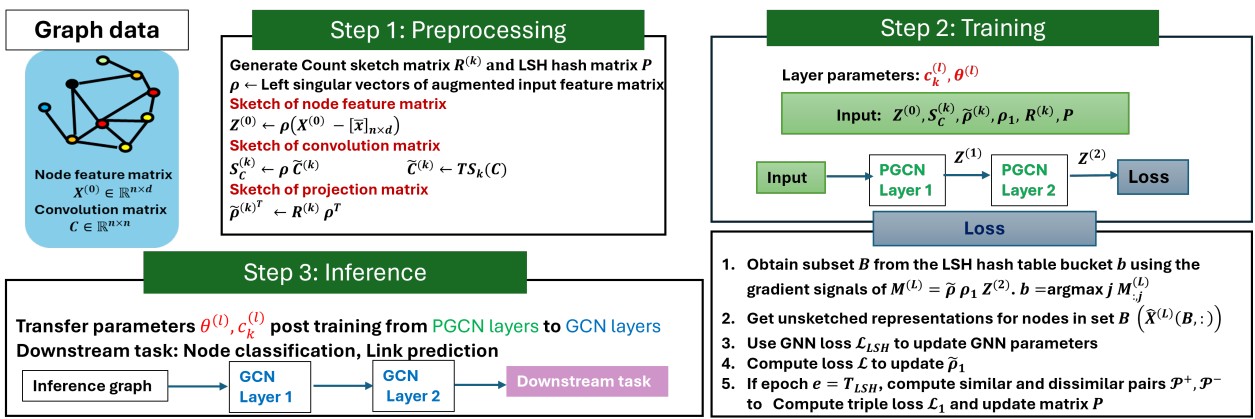

Figure 1: The preprocessing, training and inference phases of the PGNN framework are shown. In the **preprocessing step**, the sketch of the input node feature matrix $(Z^{(0)})$, $k$ sketches of the convolution matrix $(S_C^{(k)})$ and the matrix $\rho$ $(\tilde{\rho}^{(k)})$ are generated, where we make use of count-sketch matrices $R^{(k)}$. These sketch matrices are utilized in the PGCN layers for the **training step** along with the LSH hash matrix $P$. The **loss** function involves computation of GNN loss $\mathcal{L}_{LSH}$ for updating GNN parameters, loss $\mathcal{L}$ to update $\tilde{\rho}_1$, and triplet loss to update the LSH hash matrix $P$ when epoch $e = T_{LSH}$. For the **inference step**, we make use of the parameters learned but with the GCN layers as shown in the figure.

Figure 1 depicts the three-phase pipeline of our PGNN approach, encompassing preprocessing, training, and inference stages. Algorithms 1 and 2 present the complete PGNN framework, which operates within the GCN architecture. The algorithm can be generalized to the architectures discussed in sections B.2 and B.4. However, for the PGAT architecture (See section B.5), the computation of the sketch for the convolution matrix must be omitted.

---

**Algorithm 1** PGNN Training and Inference

---

**Require:** Preprocessed data: $Z^{(0)}, R^{(1)}, \ldots, R^{(k)}, S_C^k, \bar{x}, P^{(k)}$ from Algorithm 2;
          Labels $y$, LSH update interval $T_{LSH}$;
          Loss constants $\beta$, $\beta_2$, $\beta_0$, $\alpha$, $\alpha_0$, $t_+$, $t_-$ for $\mathcal{L}$ and $\mathcal{L}_1$
 1: **Training:**
 2: Initialize GNN weights $\Theta^{(l)}$, coefficients $c_k^{(l)}$, matrix $\rho_1$
 3: **for** each epoch **do**
 4:    **for** $l = 1$ to $L$ **do**
 5:       Compute $Z^{(l)}$ using the PGNN propagation rule (Eq. 14)
 6:    **end for**
 7:    Select subset $B$ ($|B| \leq \beta$) using gradient signals from LSH hash tables $H^{(k)}$ (Section 4.1.1)
 8:    Reconstruct $\hat{X}^{(L)}$ for nodes in the set $B$ (Eq. 7), evaluate loss $\mathcal{L}_{\text{LSH}}$ (Eq. 8)
 9:    Backpropagate and update $\Theta^{(l)}, c_k$ using $\mathcal{L}_{\text{LSH}}$
10:    **if** epoch mod $T_{\text{LSH}} = 0$ **then**
11:       Update matrix $P^{(k)}$ using triplet loss $\mathcal{L}_1$ (Eq. 9)
12:    **end if**
13:    Update $\rho_1$ using $\beta_2$ randomly sampled nodes using loss $\mathcal{L}$ {Omitted for PSGC}
14: **end for**
15: **return** Learned weights $\Theta^{(l)}$ and coefficients $c_k^{(l)}$
16: **Inference:**
17: Predict using standard GCN update rule with learned $\Theta^{(l)}, c_k^{(l)}$

---

Table 1: Performance comparison of PGNN with Graph-SAINT (Zeng et al., 2020), VQ-GNN (Ding et al., 2021), Sketch-GNN (Ding et al., 2022), Graph Coarsening (Cai et al., 2021) , and linearized GNN (Wu et al., 2019) on Reddit, ogbn-arxiv, and ogbn-products. If the entry is unavailable in prior literature it is denoted by '-'. The accuracy values in **green** highlight the best performance for a given method (e.g., GCN) within each column, while the values in red denote the second-best performance for that method.

| Dataset → 
 Method ↓ | ogbn-arxiv | | | Reddit | | | ogbn-products | | |
|---|---|---|---|---|---|---|---|---|---|
| SGC | 69.44 ± 0.05 | | | 94.64 ± 0.11 | | | 66.89 ± 0.29 | | |
| PSGC | 68.57 ± 0.16 | | | 94.66 ± 0.04 | | | 65.6 ± 0.00 | | |
| GNN Model | GCN | GraphSAGE | GAT | GCN | GraphSAGE | GAT | GCN | GraphSAGE | GAT |
| "Full-Graph" | 71.74 ± 0.29 | 71.49 ± 0.27 | 73.65 ± 0.11 | OOM | OOM | OOM | OOM | OOM | OOM |
| Graph-SAINT | 70.79 ± 0.57 | 69.87 ± 0.39 | 71.17 ± 0.32 | 92.25 ± 0.57 | 95.81 ± 0.57 | 94.31 ± 0.67 | 76.02 ± 0.21 | 79.08 ± 0.24 | 79.71 ± 0.42 |
| Coarsening | 68.92 ± 0.35 | 66.09 ± 0.61 | 63.07 ± 0.41 | - | - | - | - | - | - |
| VQ-GNN | 70.55 ± 0.33 | 70.28 ± 0.47 | 70.43 ± 0.34 | 93.99 ± 0.21 | 94.49 ± 0.24 | 94.38 ± 0.59 | 75.24 ± 0.32 | 78.09 ± 0.19 | 78.23 ± 0.49 |
| sketch-ratio ($r = c_0/n$) | r = 0.4 | | | r = 0.3 | | | r = 0.3 | | |
| sketch-GNN | 70.28 ± 0.87 | 70.48 ± 0.80 | 70.53 ± 0.34 | 92.80 ± 0.34 | 94.85 ± 0.61 | 93.26 ± 0.63 | 75.53 ± 1.05 | 77.62 ± 0.93 | 77.48 ± 0.71 |
| PGNN ratio ($r = c_0/n$) | r = 0.15 | | | r = 0.05 | | | r = 0.003 | | |
| PGNN | 69.53 ± 0.31 | 69.63 ± 0.08 | 70.27 ± 0.11 | 94.82 ± 0.03 | 94.32 ± 0.08 | 93.02 ± 0.09 | 75.21 ± 0.51 | 76.82 ± 0.55 | OOM |

# 5 Experiments.

We evaluate the efficiency of PGNN in terms of memory utilization and training time, with implementation details provided in Appendix E. Our assessment, conducted on benchmark graph datasets, focuses on node classification accuracy and compares PGNN's performance against state-of-the-art methods, including GCond (Jin et al., 2022) and Graph Coarsening (Cai et al., 2021). Additionally, we compare PGNN with other sampling-based methods such as GraphSAINT (Zeng et al., 2020), VQ-GNN (Ding et al., 2021), and existing sketch based method Sketch-GNN (Ding et al., 2022). For each dataset, the sketch ratios for GCond, Graph Coarsening, and Sketch-GNN remain the same, while PGNN uses either the same or lower sketch ratios than Sketch-GNN. The graph datasets used for evaluation include Cora, Citeseer, Pubmed, ogbn-arxiv, Reddit, ogbn-products and ogbn-papers100M. The PGNN update rules for various GNN architectures discussed in this section are detailed in Appendix B. Figures 2 and 3 illustrate PGNN's sublinear memory complexity and training time, making PGNN suitabile for large graph datasets. Additionally, we examine cross-architecture memory complexity in Appendix C (Figure 7). Node classification accuracies across all datasets are reported in Tables 1 and 2, demonstrating PGNN's ability to achieve competitive accuracy

Table 2: Performance comparison of PGNN with SGC, GCN, SAGE, GAT and graph compression techniques like GCond (Jin et al., 2022), Graph Coarsening (Cai et al., 2021) and Sketch-GNN (Ding et al., 2022) on Cora, Citeseer, and Pubmed datasets. The baseline graph compression ratios for the Cora, Citeseer, and Pubmed datasets are 0.026, 0.018, and 0.04, respectively, whereas our PGNN framework employs sketch ratios of $r = 0.02, 0.018$, and $0.01$ for these datasets. If the entry is unavailable in prior literature it is denoted by '-'. The accuracy values in **green** highlight the best performance for a given method (e.g., GCN) within each column, while the values in **red** denote the second-best performance for that method.

| Dataset → / Method ↓ | Cora | | | Citeseer | | | Pubmed | | |
|---|---|---|---|---|---|---|---|---|---|
| GCN | 81.19 ± 0.23 | | | 71.91 ± 0.18 | | | 79.0 ± 0.4 | | |
| SGC | 81.0 ± 0.0 | | | 71.9 ± 0.1 | | | 78.9 ± 0.0 | | |
| PSGC | 80.51 ± 0.18 | | | 72.01 ± 0.31 | | | 79.90 ± 0.10 | | |
| SAGE | 74.5 ± 0.0 | | | 67.2 ± 0.0 | | | 76.8 ± 0.0 | | |
| GAT | 83.0 ± 0.7 | | | 72.5 ± 0.7 | | | 79.0 ± 0.3 | | |
| | GCN | SAGE | GAT | GCN | SAGE | GAT | GCN | SAGE | GAT |
| Coarsening | 65.18 ± 0.51 | – | – | 59.08 ± 0.45 | – | – | – | – | – |
| GCond | 80.02 ± 0.75 | 76.18 ± 0.87 | 66.2 ± 0.0 | 70.59 ± 0.87 | 66.2 ± 0 | 55.4 ± 0 | 77.92 ± 0.42 | 71.12 ± 3.10 | – |
| Sketch-GNN | 80.35 ± 0.71 | 79.14 ± 1.21 | – | 71.14 ± 0.59 | – | – | – | – | – |
| PGNN | 80.60 ± 1.46 | 79.43 ± 1.54 | 77.82 ± 1.26 | 71.21 ± 1.13 | 68.87 ± 1.21 | 70.47 ± 4.76 | 79.23 ± 0.32 | 78.01 ± 0.18 | 78.23 ± 0.61 |

while maintaining computational efficiency. To further showcase PGNN's versatility across various tasks, we also evaluated its performance on link prediction, comparing it against established graph-based methods. The experimental results for the link prediction tasks are detailed in Table 8 in the Appendix.

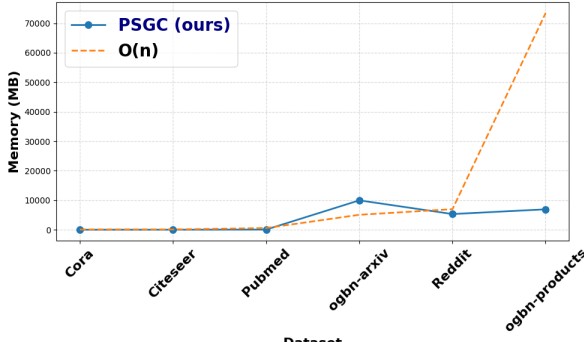

Figure 2: Memory complexity of the PSGC method.

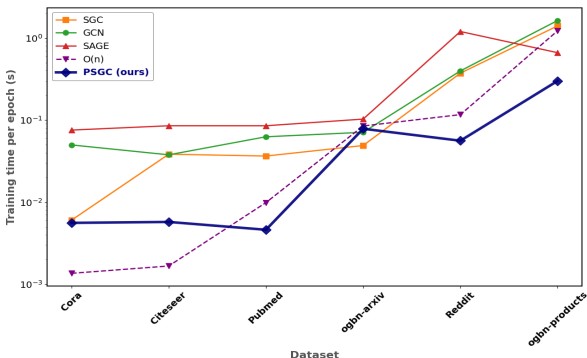

Figure 3: Training time comparison of the SGC, GCN, and SAGE with the PSGC method.

## 5.1 Performance evaluation of PGNN

As shown in Table 2, PGNN outperforms existing graph compression techniques—including Sketch-GNN, GCond, and Graph Coarsening—while closely matching the accuracy of full-graph training on the Cora, Citeseer, and Pubmed datasets, despite using lower or similar sketch ratios. For instance, PGNN achieves better classification accuracies on Cora with a sketch ratio of $r = 0.02$, compared to $r = 0.026$ for Sketch-GNN and other techniques such as GCond and Coarsening. Similarly, on Pubmed, PGNN attains superior accuracy with $r = 0.01$, compared to the $r = 0.04$ required by existing methods. Notably, PGNN maintains high classification accuracy even under lower compression levels, outperforming methods like SGC, GCN, and GraphSAGE on Pubmed (Table 2). PGNN demonstrates scalability, particularly on large-scale datasets. For the Reddit dataset, PGNN achieves an accuracy of 94.82% using a sketch ratio of only 0.05, compared to Sketch-GNN, which requires a sketch ratio of 0.3 to reach a lower accuracy of 92.0%. PGNN with SGC architecture achieves better classification accuracies than SGC. Additionally, PGNN's **memory footprint** for the Reddit dataset remains as low as 5400 MB with PSGC and approximately 7500 MB with PGCN (Figure 7). For the ogbn-products dataset we observe that with a low sketch-ratio of 0.003 we are able to reach full-graph training accuracy for the GCN architecture unlike Sketch-GNN which uses a much higher sketch-ratio of 0.3. PGCN consistently outperforms GCN architectures in link prediction tasks in the four

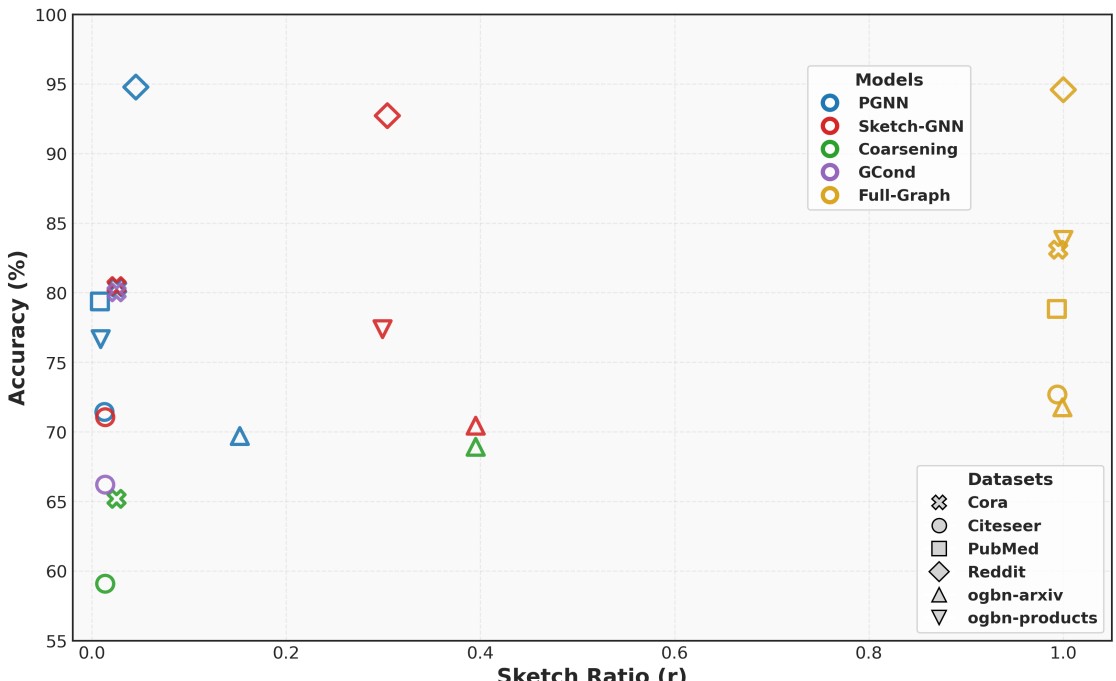

Figure 4: Performance comparison of different models across varying sketch ratios. PGNN (blue) achieves comparable accuracy to other methods while maintaining lower sketch ratios, demonstrating its efficiency in preserving accuracy with reduced computational complexity. The comparison includes Sketch-GNN (red), Graph Coarsening (green), GCond (purple), and the Full-Graph (orange) baseline accuracies.

datasets: Core, Citeseer, Pubmed, and ogbn-arxiv, as shown in Table 8. Regarding **training times**, PGNN demonstrates notable efficiency: the training time per epoch for PSGC is consistently lower than that of SGC (except on ogbn-arxiv) and outperforms both GCN and GraphSAGE across all evaluated datasets (Figure 3). Unlike Sketch-GNN, PGNN does not require updating LSH hash tables at intermediate layers, thereby reducing computational overhead.

In terms of **preprocessing**, methods such as GCond and Graph Coarsening, while effective for small-scale graphs, exhibit scalability limitations. For instance, preprocessing the Reddit dataset takes approximately 90 minutes with GCond, whereas PGNN completes the preprocessing in around 560 seconds (Table 4). Moreover, PGNN requires a lower sketch ratio to maintain full-graph accuracy as the graph size increases, underscoring its adaptability to larger datasets (Figure 4). Despite processing a massive dataset of 111 million nodes, our approach achieves preprocessing in just 28 minutes.

PGCN on ogbn-arxiv incurs a 2.5% accuracy drop compared to full-graph GCN (Table 1). However despite using a sketch ratio of $r = 0.15$ compared to $r = 0.4$ for baseline methods we observe only a 1% accuracy drop with respect to existing graph compression techniques. Despite PGNN's preprocessing time not being linear, it remains approximately one-sixth of the time required by graph compression algorithms such as GCond (Jin et al., 2022), Graph Coarsening (Cai et al., 2021) (Table 4). The out-of-memory (OOM) error for PGAT on ogbn-products at a low sketch ratio of 0.003 occurs because GAT's learnable convolution mechanism cannot leverage convolution matrix sketching (Appendix B.5), thus preventing the memory advantages achieved by other PGNN architectures.

## 5.2 ogbn-papers100M: Paper Citation Network

We evaluate our PGNN framework on ogbn-papers100M, a large-scale citation network containing 111 million nodes. This dataset follows the same preprocessing methodology as ogbn-arxiv Wang et al. (2019), ensuring consistency in feature construction and graph representation.

Due to computational constraints, baseline results for SGC Wu et al. (2019), Node2Vec Grover & Leskovec (2016), and MLP are obtained from the OGB leaderboard Hu et al. (2020), as full-batch training of these methods requires more than 512 GB of CPU memory. Our experiments use a sketch ratio of $10^{-6}$ for the ogbn-papers100M dataset, selected based on available memory limitations. To enable a fair comparison of memory usage, we keep the batch size to 1024 for both GCN and GraphSAGE models.

Table 3 compares PGNN with methods like SGC, GCN, GraphSAGE, MLP, Node2Vec for the ogbn-papers100M dataset. The preprocessing overhead for the ogbn-papers100M dataset requires only 29 minutes, as shown in Table 4. PSGC significantly enhances memory efficiency and training speed while maintaining competitive accuracy. It outperforms MLP and Node2Vec, and achieves only a 4% accuracy reduction compared to the full SGC model. Notably, it achieves a $339\times$ speedup for the training time per epoch compared to the conventional models. For PSGC, PGCN, and PSAGE, we utilized a POD sketch ratio of $10^{-6}$ ($r = 10^{-6}$). The tensor-sketch ratio for PGCN was set to the same value. However, for PSAGE, a tensor-sketch ratio of $5 \times 10^{-7}$ was used. Higher ratios were not feasible due to CPU hardware limitations for preprocessing.

For PGCN and PSAGE on ogbn-papers100M, we utilise PGNN layers during inference.

Table 3: Performance evaluation of **PGNN** against established graph-based and traditional methods (SGC Wu et al. (2019), GCN Kipf & Welling (2017), GraphSAGE Hamilton et al. (2018), Node2Vec, and MLP) on the large-scale ogbn-papers100M dataset. Blue: PGNN method; green: best memory/training time; red: Out of Memory (OOM); orange: Speedup factor.

| Method | Test Accuracy (%) | Memory (GB) | Training time per epoch (s) | Speedup |
|---|---|---|---|---|
| SGC | $63.29 \pm 0.19$ | $> 512$ | **OOM** | — |
| **PSGC (ours)** | $59.42 \pm 0.01$ | **7.70** | **0.62** | — |
| MLP | $55.60 \pm 0.23$ | $> 512$ | **OOM** | — |
| GCN | $27.18 \pm 0.01$ | 10.42 | 221.79 | — |
| **PGCN (ours)** | $56.35 \pm 0.00$ | **7.72** | **0.65** | $\sim \mathbf{339\times}$ |
| Node2Vec | $47.24 \pm 0.31$ | $> 512$ | **OOM** | — |
| GraphSAGE | $67.06 \pm 0.17$ | 10.53 | 223.82 | — |
| **PSAGE (ours)** | $50.78 \pm 1.37$ | **3.58** | **0.63** | $\sim \mathbf{339\times}$ |

## 6 Conclusions

The computational and memory demands of large-scale graph learning pose significant challenges for modern GNN frameworks. To address this, we propose PGNN, a novel sketch-based framework that compresses graph data while preserving the downstream task performance. Our experiments demonstrate that PGNN achieves competitive accuracy with state-of-the-art methods at significantly reduced sketch ratios and sublinear memory complexity. The condensed representations generated by PGNN not only reduce storage overhead but also enable efficient training across diverse GNN architectures. Future work include (1) broadening PGNN to dynamic graphs and streaming scenarios, and (2) extending the framework to heterogeneous GNN architectures.

## Acknowledgements

We thank the anonymous reviewers of TMLR, the Action Editor, and the Editors-in-Chief for their constructive feedback, which helped improve the paper.

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

## A    Appendix

**Definition 1.** *If the column space of matrix $B$ is a subset of matrix $A$ there exists a matrix $M$ such that $B = AM$.*

Before establishing Theorem 1, we clarify an assumption central to its proof. Lemma 1 captures a key column-space invariance property: whenever $\mathrm{col}(A) \subseteq \mathrm{col}(B)$, a matrix $M$ exists (and is uniquely determined) such that $A = BM$. This observation, along with the conditions ensuring consistency of the associated linear system, is fundamental for the argument of Theorem 1.

**Lemma 1.** *If the column space of $B = C^{(l)}\rho^T$ is contained in the column space of $\rho^T$, then the matrix $M$ defined in Definition 1 takes the form*

$$M = \rho B.$$

*proof:* By assumption,

$$\mathrm{col}(B) \subseteq \mathrm{col}(\rho^T).$$

Hence, there exists a matrix $M$ such that

$$B = \rho^T M \quad \text{(Definition 1)}.$$

Left-multiplying both sides by $\rho$ and using the orthonormality condition $\rho\rho^T = I_{c_0}$ yields

$$M = \rho B.$$

Thus $M = \rho B$ is the unique matrix satisfying $C^{(l)}\rho^T = \rho^T M$.
In what follows, we demonstrate that there indeed exist vectors fulfilling this condition.
**Existence of solution.** To ensure the existence of a vector $z \in \mathbb{R}^{c_0}$ satisfying

$$C^{(l)}\rho^T z = \rho^T M z = \rho^T \rho C^{(l)}\rho^T z = P C^{(l)}\rho^T z,$$

the corresponding linear system must be consistent. Since this system is overdetermined, $z$ must lie in the nullspace of $C^{(l)}\rho^T$. This leads to the requirement

$$C^{(l)}\rho^T z = \mathbf{0}_n. \tag{12}$$

When $C = I - D^{-1/2}AD^{-1/2}$, the normalized Laplacian Chung (1997) with degree matrix $D$ and adjacency matrix $A$, we have:

$$\mathrm{span}(D^{1/2}\mathbb{1}_n) \in \mathrm{null}(C).$$

We also have

$$\mathrm{span}(D^{1/2}\mathbb{1}_n) \in \mathrm{null}(C^{(l)}).$$

Therefore, if $\rho^T z \in \mathrm{span}(D^{1/2}\mathbb{1}_n)$, then equation 12 holds, and since the column space of $\rho^T$ coincides with that of $F = \frac{\hat{X}^{(0)} - [\bar{x}]_{n \times c_0}}{\sqrt{nc_0 - 1}}$, it suffices that one or more columns of $F$ be scalar multiples of $D^{1/2}\mathbb{1}_n$ to ensure the existence of such a vector $z$.

### A.1    Proof for Theorem 1

The output layer node representations for the linearized GNN using the optimal approximation of the input node feature matrix given by the POD,

$$\hat{X}^{(l)} = C^{(l)}\rho^T Z^{(0)}\Theta + [C^{(l)}\bar{x}]_{n \times d}\Theta = B\Psi + \Psi_1,$$

$B = C^{(l)}\rho^T, \Psi = Z^{(0)}\Theta, \Psi_1 = [C^{(l)}\bar{x}]\Theta$. Consider $d$ as the number of hidden channels at layer $l$. We propose that the projection matrix, denoted by $Q$, is the product of two matrices $Q_1$ and $Q_2$, such that $Q = Q_1 Q_2$. In this context, $Q_2$ signifies the linear transformation responsible for reducing the dimensionality of matrices.

Conversely, $Q_1$ is the matrix that handles the inverse projection. The optimal projection matrix $Q^*$ is obtained from solving the optimization problem described below,

$$\min_{Q \in \mathcal{P}} \|QB\Psi - B\Psi\|_F^2 = \min_{Q \in \mathcal{P}} \sum_{i=1}^{c_0} \|QB\Psi(:,i) - B\Psi(:,i)\|_2^2$$

When the columnspace of matrix $B = C\rho^T$ is assumed as a subset of columnspace of matrix $\rho^T$, then for every vector $B\Psi(:,i)$, there exists $\alpha_i = \rho B\Psi(:,i)$ such that $B\Psi(:,i) = \rho^T \alpha_i$ (Lemma 1). Also since the matrix $\rho^T$ denote the left singular vectors of the SVD of the matrix $F = \frac{\tilde{X}^{(0)} - [\bar{x}]_{n \times c_0}}{\sqrt{nc_0 - 1}}$, $\text{col}(F) = \text{span}(\rho^T(:,1), \rho^T(:,2), \ldots \rho^T(:,c_0))$. Thus there exist $\omega_i \in \mathbb{R}^{c_0}$ such that $\rho^T \alpha_i = F\omega_i$. The matrix $\tilde{X}^{(0)}$ denotes the augmented input node feature matrix (Algorithm 2). The optimization problem in equation A.1 becomes

$$\min_{Q \in \mathcal{P}} \sum_{i=1}^{c_0} \|QF\omega_i - F\omega_i\|_2^2 = \min_{Q \in \mathcal{P}} \sum_{i=1}^{c_0} \|(QF - F)\omega_i\|_2^2$$

$$\leq \min_{Q \in \mathcal{P}} \|QF - F\|_F^2 \sum_{i=1}^{c_0} \|\omega_i\|_2^2, \qquad \text{(since } \|Ax\|_2 \leq \|A\|_2\|x\|_2 \leq \|A\|_F\|x\|_2). \tag{13}$$

Given a dataset $X$, the POD projection matrix $P_1^*$ minimizes

$$e(P_1, X) = \langle P_1 X - X, \, P_1 X - X \rangle \quad \text{(see Rathinam \& Petzold (2003))}.$$

This property remains valid when the dataset is centered to obtain $F$. Therefore, the matrix $\tilde{Q}^*$ which minimizes the R.H.S of inequality 13 is the POD projection matrix $P$ corresponding the centered dataset $F$.

## A.2 Proof for Theorem 2

Let the equivalent node representations at layer $l+1$ in the graph dimension $n$, be $\hat{X}^{(l+1)}$, the PSGC update rule (Appendix B.2) is given by the following expression:

$$\hat{X}^{(l+1)} = PC\hat{X}^{(l)}\Theta^{(l)} + (I - P)[\bar{x}]_{n \times d_{(l+1)}},$$

$$\left\| X^{(l+1)} - \hat{X}^{(l+1)} \right\| = \left\| CX^{(l)}\Theta^{(l)} - (PC\hat{X}^{(l)}\Theta^{(l)} + (I - P)[\bar{x}]_{n \times d_{(l+1)}}) \right\|,$$

$$\left\| X^{(l+1)} - \hat{X}^{(l+1)} \right\| = \left\| CX^{(l)}\Theta^{(l)} - C_{eq}X^{(l)}\Theta^{(l)} + C_{eq}X^{(l)}\Theta^{(l)} - (C_{eq}\hat{X}^{(l)}\Theta^{(l)} + (I - P)[\bar{x}]_{n \times d_{(l+1)}}) \right\|,$$

Using the triangular inequality and sub-multiplicative property of norms,

$$\epsilon^{(l+1)} \leq \frac{\|C - C_{eq}\| \, \|\Theta^{(l)}\|}{S} + \frac{\epsilon^{(l)} \, \|C_{eq}\| \, \|\Theta^{(l)}\|}{S} + \bar{T}, \;\; S = \frac{\|X^{(l+1)}\|}{\|X^{(l)}\|}.$$

**Lemma 2** (Ding et al. (2022)). *Given matrices $C \in \mathbb{R}^{n \times n}$ and $(X^{(l)}W^{(l)})^T \in \mathbb{R}^{d \times n}$, consider a randomly selected count sketch matrix $R \in \mathbb{R}^{c_k \times n}$ (defined in section 3.1), where $c_k$ is the sketch dimension, and it is formed using $r = \sqrt{jn}$ underlying hash functions drawn from a 3-wise independent hash family $\mathcal{H}$ for some $j \geq 1$. If $c_k \geq \frac{2 + 3j}{\varepsilon^2 \delta}$, we have*

$$\Pr\{\|(CR_k^T)(R_k X^{(l)}W^{(l)}) - CX^{(l)}W^{(l)}\|_F^2 > \varepsilon^2 k \|C\|_F^2 \|X^{(l)}W^{(l)}\|_F^2\} \leq \delta.$$

# B Generalize to more GNNs

This section presents a compendium of prevalent GNNs that can be tailored to fit into the unified framework delineated in section 3. The crux of most GNN architectures revolves around message passing among node features, followed by feature transformation and activation functions—a process commonly known as 'generalized graph convolution'. Within this overarching framework, the distinctions among GNNs primarily

arise from their choice of convolution matrices, denoted as $C^{(q)}$, which can either remain static or evolve as trainable parameters. A trainable convolution matrix is contingent upon input data and adjustable parameters, potentially varying across different layers, as denoted by $C(l, q)$.

$$C_{i,j}^{(l,q)} = \underbrace{\mathcal{C}_{i,j}^{(l,q)}}_{\text{fixed}} \cdot \underbrace{h_{\theta^{(l,q)}}^{(q)}(X_{i,:}^{(l)}, X_{j,:}^{(l)})}_{\text{learnable}}$$

### B.1 Theoretical Guarantees for nonlinear GNN architectures.

We analyze how the PGNN framework extends to various nonlinear GNN architectures, including GCN (Kipf & Welling, 2017), GraphSAGE (Hamilton et al., 2018), and GAT (Veličković et al., 2018).While Theorem 1 establishes PGNN's optimality specifically for a projection subproblem induced by the linearized GNN case (SGC), the framework maintains rigorous performance guarantees for nonlinear architectures through general sketch-based approximation theory. To quantify the approximation quality for these nonlinear cases, we adapt error bound results from Ding et al. (2021) as Theorems 3 and 4, which provide bounds for forward-pass node representations and back-propagated gradients, respectively. This theoretical framework ensures reliable approximation quality across diverse message-passing schemes and provides crucial insights into how compressed computations affect the learning process relative to standard GNN update rules.

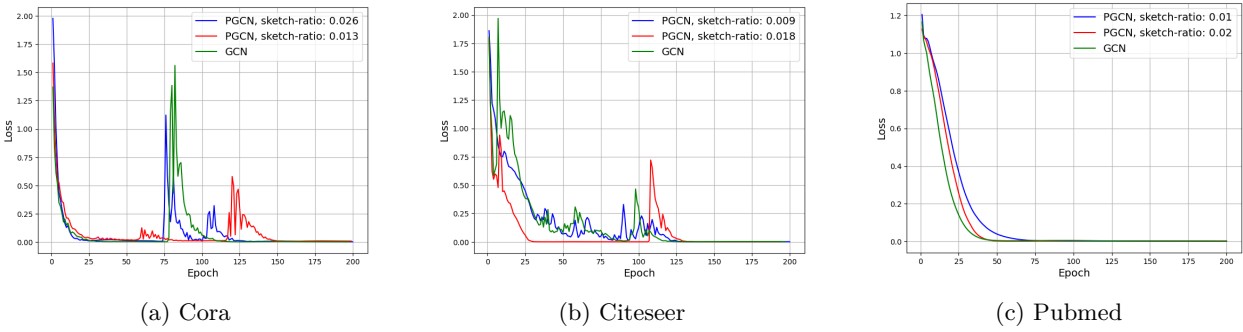

(a) Cora  (b) Citeseer  (c) Pubmed

Figure 5: Comparison of training losses during link prediction for PGCN and GCN methods across benchmark datasets: (a) Cora, (b) CiteSeer, and (c) PubMed. The sketch ratios used are 0.013 and 0.026 for Cora, 0.009 and 0.018 for CiteSeer, and 0.01 and 0.02 for PubMed, respectively.

**Theorem 3** (Ding et al. (2021)). *If the relative error of the l-th layer for the PGNN method is $\varepsilon^{(l)}$, the convolution matrix $C^{(l)}$ is either fixed or learnable with the Lipschitz constant of $h_\theta^{(l)}(\cdot) : \mathbb{R}^{2f_l} \to \mathbb{R}$ upper-bounded by $Lip(h_\theta^{(l)})$, and the Lipschitz constant of the non-linearity is $Lip(\sigma)$, then the estimation error of forward-passed features satisfies,*

$$\|\hat{X}^{(l+1)} - X^{(l+1)}\|_F \le \varepsilon^{(l)} \cdot (1 + O(Lip(h_\theta^{(l)})))Lip(\sigma)\|C^{(l)}\|_F\|X^{(l)}\|_F\|W^{(l)}\|_F.$$

**Theorem 4** (Ding et al. (2021)). *If the conditions in Theorem 3 hold and the non-linearity satisfies $|\sigma'(z)| \le \sigma'_{max}$ for any $z \in \mathbb{R}$, then the estimation error of back-propagated gradients satisfies,*

$$\|\hat{\nabla}_{X^{(l)}}\ell - \nabla_{X^{(l)}}\ell\|_F \le \varepsilon^{(l)} \cdot (1 + O(Lip(h_\theta^{(l)}))\sigma'_{max}\|C^{(l)}\|_F\|\nabla X^{(l+1)}\|_F\|W^{(l)}\|_F.$$

In the below subsections, we discuss how the PGNN framework works with various GNN architectures using update rules as described in the supplementary material of Ding et al. (2022).

## B.2 PGNN with SGC.

The convolution matrix for PSGC, $C = \tilde{D}^{-1/2}\tilde{A}\tilde{D}^{-1/2}$. $\tilde{D} = D + I_n, \tilde{A} = A + I_n$. $D$ denotes the degree matrix and $A$ represents the adjacency matrix of the graph. The node representation at layer $l + 1$, $Z^{(l+1)}$ given by the PGNN method is $Z^{(l+1)} = S_C Z^{(l)}\Theta^{(l)} + [\rho C\bar{x}]_{n \times d_l}\Theta^{(l)}$. The sketch of the convolution matrix $S_C = \rho C \rho^T$.

## B.3 PGNN with GCN.

The update rule for PGCN is given by

$$Z^{(l+1)} = \sum_{k=1}^{q} c_k \, \rho_1 \, S_C^k \left[ \text{FFT}^{-1} \left( \bigodot_{p=1}^{k} \text{FFT}\left( V_{\tilde{N}}^{(p)} \right) \right) \right]^T - U \tag{14}$$

$U = [\rho\bar{x}]_{c_0 \times d_{l+1}}$. The convolution matrix $C$ has the same form as the PSGC update rule (Appendix B.2). $S_C^k = \rho \, \text{TS}_k(C)$. Tensor-sketch of order $k$, $\text{TS}_k(C) = \left[ \text{FFT}^{-1} \left( \bigodot_{p=1}^{k} \text{FFT}\left( \text{CS}^{(p)}(C) \right) \right) \right]$. $\tilde{N} = (\rho^T \rho_1 Z^{(l)} + [\bar{x}]_{n \times d_l})\Theta^{(l)}$, $V_{\tilde{N}}^{(p)} = \text{CS}^{(p)}(\tilde{N}^T) = \Theta^{(l)^T}\left( [\bar{x}^T R^{(k)^T}]_{d_l \times c_k} + Z^{(l)^T}\rho_1 \tilde{\rho} \right)$, $\tilde{\rho} = \rho R^{(k)^T}$, $R^{(k)} \in \mathbb{R}^{n \times c_k}$ denotes the Count-sketch matrix (Section 3). Detailed steps for the PGCN method are explained in Algorithm 1.

## B.4 PGNN with SAGE.

We extend PGNN to architectures involving multiple convolutions $C^{(1)} = I_{n \times n}$ and $C^{(2)} = D^{-1}A$. $C = \left[ C^{(1)} \| C^{(2)T} \right]^{\top}$. The update rule for SAGE is

$$X^{(l+1)} = \sigma\left( X^{(l)}W^{(l,1)} + D^{-1}AX^{(l)}\Theta^{(l,2)} \right) = \sigma\left( \left[ I_n \| (D^{-1}A)^{\top} \right]^{\top} \left[ X^{(l)}\Theta^{(l,1)} \| X^{(l)}\Theta^{(l,2)} \right] \right). \tag{15}$$

With $U = [\rho\bar{x}]_{n \times d_{l+1}}$. The update rule for PSAGE is given by

$$Z^{(l+1)} = \sum_{k=1}^{q} c_k \, \rho_1 \, S_C^k \left[ \text{FFT}^{-1} \left( \bigodot_{p=1}^{k} \text{FFT}\left( V_{\tilde{N}}^{(p)} \right) \right) \right]^T - U.$$

The sketch of the convolution matrix $S_C^k = \rho \, \text{TS}_k(C)$. Tensor-sketch of order $k$,

$$\text{TS}_k(C) = \text{FFT}^{-1} \left( \bigodot_{p=1}^{k} \text{FFT}\left( \text{CS}^{(p)}(C) \right) \right), \text{CS}^{(p)}(C) = CR^{(p)\top}, \quad R^{(p)} \in \mathbb{R}^{2n \times c_k}.$$

$$\tilde{N} = \left[ \left( \rho^{\top}\rho_1 Z^{(l)} + [\bar{x}]_{n \times d_l} \right)\Theta^{(l,1)} \| \left( \rho^{\top}\rho_1 Z^{(l)} + [\bar{x}]_{n \times d_l} \right)\Theta^{(l,2)} \right].$$

$V_{\tilde{N}}^{(p)} = \text{CS}^{(p)}(\tilde{N}^{\top}) = \tilde{N}^T R^{(p)\top} = \left[ \left( \tilde{\rho}^{(1,p)^{\top}}\rho_1 Z^{(l)} + U_1 \right)\Theta^{(l,1)} \| \left( \tilde{\rho}^{(2,p)^{\top}}\rho_1 Z^{(l)} + U_2 \right)\Theta^{(l,2)} \right]^{\top}$. $\tilde{\rho}^{(1,p)} = \rho R^{(p)}(1 : n, :)$, $\tilde{\rho}^{(2,p)} = \rho R^{(p)}(n + 1 : 2n, :)$, $U_1 = \left[ R^{(p)\top}(:, 1 : n)\bar{x} \right]_{c_k \times d_l}$, $U_2 = \left[ R^{(p)\top}(:, n + 1 : 2n)\bar{x} \right]_{c_k \times d_l}$.

## B.5 PGNN with GAT.

The convolution mechanism intrinsic to the GAT architecture is inherently learnable. We propose PGAT update rule for update rules involving learnable convolution for completeness. There is however no memory or training advantage when using PGAT. A promising direction for future work lies in designing GAT update rules that leverage the PGNN method's computational advantages. Equation 6 defines the **unsketch**

operation.

$$U = [\rho \bar{x}]_{n \times d_{l+1}}, \quad \tilde{E}^{(l,q)} = F^{(l,q)} + F^{(l,q)^T}$$

$$F^{(l,q)} = \mathbf{unsketch}\left(Z^{(l,q)}\right)\Theta^{(l,q)}$$

$$a^{(l,q)} = (\rho^T \rho_1 Z^{(l,q)} + [\bar{x}]_{n \times d_l})\Theta^{(l,q)}a^{(l,q)}$$

$$C = A + I, \quad a^{(l,q)} \in \mathbb{R}^{d_{l+1}}$$

$$C^{GAT} = C \odot \exp(\text{LeakyReLU}(\tilde{E}^{(l,q)})) \tag{16}$$

$$Z^{(l+1,q)} = \rho_1 \rho \, \sigma\left(\mathbf{softmax}\left(C^{GAT}\right)\mathbf{unsketch}\left(Z^{(l,q)}\right)\Theta^{(l,q)}\right) - U \tag{17}$$

---

**Algorithm 2** PGNN Preprocessing

---

**Require:** Node feature matrix $X^{(0)} \in \mathbb{R}^{n \times d}$, convolution matrix $C$, sketch ratio $r = \frac{c_0}{n}$, number of sketches $k$, count sketch matrix dimension $c_k$

1: Set $c_0 = \lceil rn \rceil$
2: **if** $c_0 > d$ **then**
3:     $r_1, r_2 \leftarrow$ randomly sample $n$ indices from $\{1, 2, \ldots, d\}$ with replacement
4:     $M = X^{(0)}(:, r_1) \odot X^{(0)}(:, r_2)$
5:     Augmented node feature matrix: $\tilde{X}^{(0)} = [X^{(0)}, M]$
6:     $\tilde{X}^{(0)} \leftarrow C\tilde{X}^{(0)}$ {Omitted for PSGC}
7: **else**
8:     Set $\tilde{X}^{(0)} = X^{(0)}$
9: **end if**
10: Compute mean vector $\bar{x} \leftarrow \frac{1}{c_0} \sum_{i=1}^{c_0} \tilde{X}^{(0)}(:, i)$
11: $\rho^T \leftarrow$ left singular vectors from SVD of $\sigma\left(\frac{\tilde{X}^{(0)} - [\bar{x}]_{n \times c_0}}{\sqrt{c_0 n - 1}}\right)$
12: Sketch of input node feature matrix: $Z^{(0)} = \rho(X^{(0)} - [\bar{x}])$
13: Generate count-sketch matrices $R^{(1)}, \ldots, R^{(k)}$
14: Compute sketch $S_C^k = \rho \, \text{TS}_k(C)$ {Omitted for PGAT}
15: Compute auxiliary terms for forward propagation: $\rho\bar{x}$, $\bar{x}^\top R^{(k)^\top}$, and $\tilde{\rho}^{(k)} = \rho R^{(k)^\top}$
16: Initialize LSH projections $P^{(k)}$
17: **return** Preprocessed sketch $Z^{(0)}$, count-sketch matrices $R^{(1)}, \ldots, R^{(k)}$, sketch $S_C^k$, $\bar{x}$, LSH projections $P^{(k)}$

---

## C  Complexity analysis

We delineate the intricacies inherent in the algorithm with the PGNN framework.

**One-time Preprocessing:** The pre-processing step involves finding the right singular vectors of the matrix described in Algorithm 2, which takes time $O(dnc_0)$. Computing $S_C = \rho C \rho^T$ for PSGC takes $O(n^2)$ computations and $S_C = \rho \text{TS}_k(C)$ for PGCN, PSAGE takes $O(c_0 c_k n)$ computations. Computing the sketch of the initial node feature matrix $S_X = \rho(X^{(0)} - [\bar{x}]_{n \times d})$ takes time $O(nc_0 d)$. Computing the sketches of the matrix $\rho$ to obtain $\tilde{\rho}$ has linear time complexity.

**Overhead of computing LSH hash tables.** The time complexity for computing the hash index for each node is $O(c_0 c_k)$ when using Simhash (See section 3), and since there are $n$ nodes and $f$ hash tables, we get an overhead of $O(fnc_0 c_k)$ for time and $O(fc_0 c_k)$ for space.

Overall, the preprocessing phase has a time complexity of $O(n^2)$ and a memory complexity of $O(n)$ for PSGC. $O(fc_0 c_k n)$ time complexity and $O(n)$ memory complexity for PGCN, PSAGE. The time consuming part involved in the preprocessing phase is the formation of matrix $\rho$, we present the time consumed by this

process in Table 4

**Training complexities with PGNN.** We present the complexities within the context of the GCN architecture. Forward and backward pass: Computing $V_{\tilde{N}}^{(p)}$ involves $O(dc_0^2) + O(dc_0 c_k)$. FFT and inverse FFT involves $O(dc_k \log(c_k))$. This reduces to $O(dc_0^2) + (dc_0 c_k)$. The memory complexity involved is $O(c_0^2) + O(c_0 c_k)$.

**Complexity associated with loss $\mathcal{L}_{LSH}$.**

1. Computing the subset $B$ of nodes based on gradient signals involves $O(c_k)$ computations. Unsketching $\beta$ number of node representations involve $O(\beta c_k d)$ computations.

2. $O(\beta d^2)$ for computing similar and dissimilar node pairs. Computing and updating hash table $P^{(k)}$ using $\mathcal{L}_1$ will involve $O(c_k d)$ operations, which needs to be updated every $T_{epoch}$ epoch.

**Complexity associated with loss $\mathcal{L}_1$ (Section 4.1.1).** Computing loss and updating $\mathcal{L}_1$ requires $O(\beta_2 c_k d)$ computations.

**Inference**: incurs $O(Ld(\frac{m}{n} + d))$ time and $O(m + ndL + d^2 L)$ memory as is the case in a standard GCN.

**Remark.** The training complexities mentioned above do not hold for the GAT architecture (Veličković et al., 2018) because of the inherent nature of the operations involved, which is expounded in Appendix B.5. The underlying complexities in the original GAT architecture will hold, and for completeness, we present the accuracies for the various datasets using PGAT in Tables 1, 2. **An implementation detail.** When

Table 4: Computation time ($\rho$) for different datasets at specific sketch ratios ($r = \frac{c_0}{n}$).

| Dataset | Sketch-ratio $r = \frac{c_0}{n}$ | $\rho$ (Time in s) |
|---|---|---|
| Cora | 0.02 | 0.0141 |
| Citeseer | 0.018 | 0.0254 |
| Pubmed | 0.01 | 0.1558 |
| ogbn-arxiv | 0.15 | 1786.8692 |
| Reddit | 0.05 | 426.9778 |
| ogbn-products | 0.003 | 350.93 |
| ogbn-papers100M | 1e-6 | 1708.40 |

the sketch ratio $r$ is such that $\lceil rn \rceil > d$, which is the feature dimension, the PCA or the POD method necessitates computing the covariance matrix (Ding et al., 2021). To overcome the challenge of storing and computing the covariance matrix, we use the feature engineering method to augment $X^{(0)}$ by selecting random combinations of columns of this matrix to find the augmented input node feature matrix $\tilde{X}^{(0)}$ (See Algorithm 2).

**Preprocessing complexity scaling analysis.** The dominant factor in preprocessing involving computation of the matrix $\rho$ of projection as reported in Table 4 exhibits distinct scaling behaviours across different graph size regimes, as illustrated in Figure 6. We analyze these patterns by comparing observed runtime against theoretical bounds.

**Small-Scale Regime ($10^3$–$10^4$ nodes):** For citation networks including Cora, Citeseer, and Pubmed, the preprocessing time scales consistently with $O(n^{1.5})$, closely tracking the theoretical reference line in our complexity plot. This near-optimal scaling demonstrates the effectiveness of POD-based sketching for modestly-sized graphs with standard feature dimensions ($\leq 128$).

**Intermediate-Scale Regime ($10^5$–$10^6$ nodes):** For Reddit and ogbn-arxiv, we observe performance degradation approaching $O(n^2)$ scaling, indicating a "scaling bottleneck" in this regime. This behaviour stems from two factors: Reddit's higher feature dimensionality (602 features) increases the $O(dnc_0)$ complexity term, while ogbn-arxiv's elevated sketch ratio similarly amplifies this cost, where $d$ represents the feature dimension.

**Large-Scale Regime ($10^6+$ nodes):** Performance recovers dramatically at scale. The ogbn-products dataset (2.4M nodes) returns to near-$O(n^{1.5})$ scaling, while ogbn-papers100M (111M nodes) achieves sub-$O(n^{1.5})$ performance, falling below the theoretical reference line. This super-linear efficiency at massive scale likely results from a lower sketch-ratio scaling compared to the graph dimension $n$.

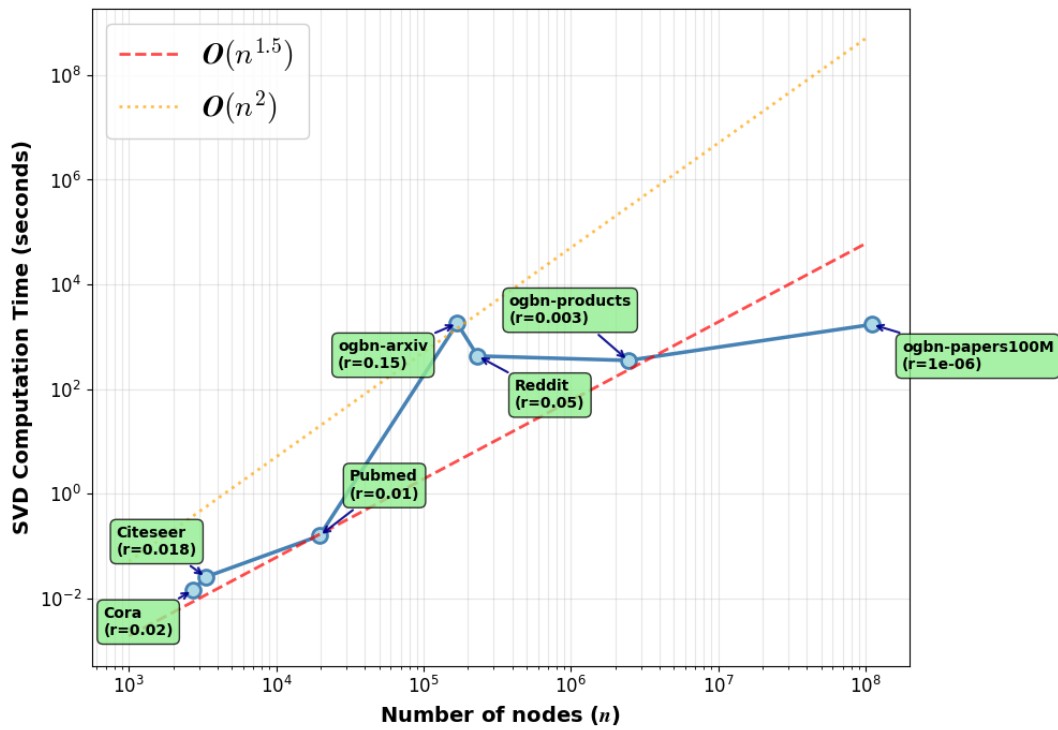

Figure 6: Preprocessing complexity scaling analysis.

## C.1 Experimental Evaluation of Memory Requirements for PGNN across architectures

Figure 7 compares the memory consumption of our proposed methods—PSGC, PSAGE, and PGCN—across various datasets. The results demonstrate that PSGC achieves the lowest memory usage. The tensor-sketch ratio used for cora, citeseer, ogbn-arxiv and reddit are 0.7, 0.5, 0.15, 0.15 with order $q$ and number of sketches $k$ as 3 (Algorithm 1). These experiments were conducted without mini-batching. PSAGE incurs nearly twice the memory complexity of PGCN due to the transfer of two convolution matrices to GPU memory.

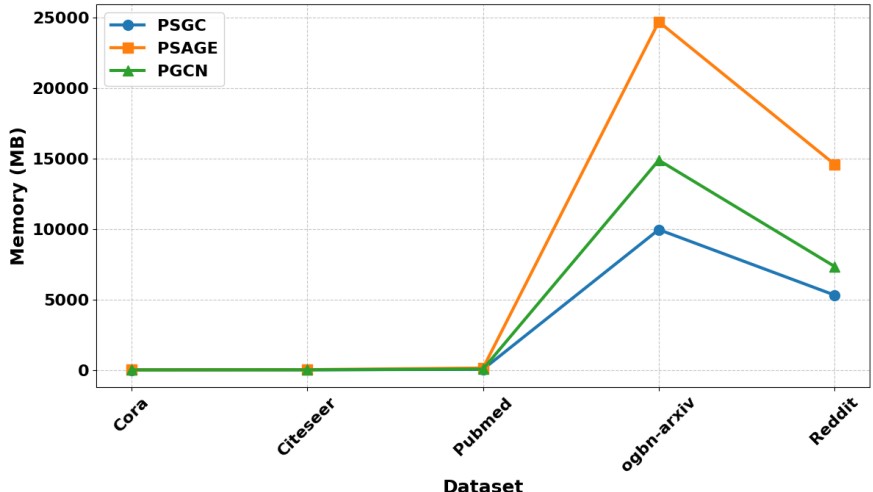

Figure 7: Memory consumption comparison of PSGC, PSAGE, and PGCN across various datasets.

# D Additional Experiments

## D.1 Comparing performance improvements obtained when using jumping knowledge networks.

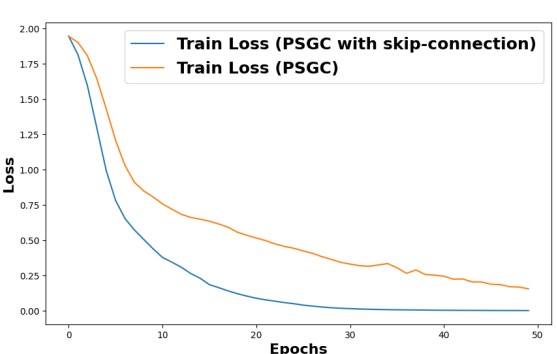

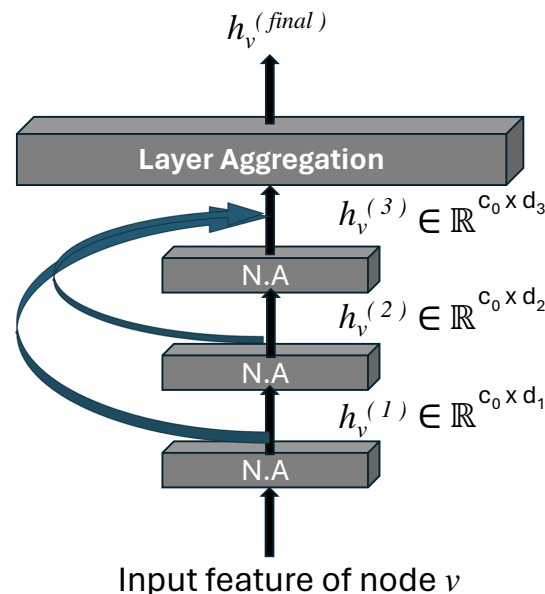

Figure 8: Classification loss when using Jumping knowledge network architecture on the PSGC versus the PSGC on a 3-layer GNN on the Cora dataset.

Figure 9: Jumping Knowledge network architecture for the PGNN method. N.A. denotes neighbourhood aggregation.

As the depth of GNNs increases, there is a tendency for the node representations to converge to a standard value, a phenomenon called "over-smoothing" (Li et al., 2018). A widely adopted mitigation approach in the literature is to bypass intermediate layers and directly contribute to the future layers by combining the Jumping Knowledge framework with models like GCN and GraphSAGE. The Jumping Knowledge (JK) framework (Xu et al., 2018; Sun et al., 2024) aggregates features from multiple GNN layers, enhancing expressiveness and robustness while addressing the issue of oversmoothing. In PGNN, as the depth of the GNNs increases, there is an accumulation of error, as shown for the linearized GNN in Theorem 2 affecting the downstream task. We use the skip-connections in the Jumping Knowledge framework as shown in Figure 9 while presenting the classification loss for the convergence aspect in Figure 8. Empirically, we find that the loss in accuracy due to depth for the Cora dataset was compensated by introducing skip connections as described in the Jumping Knowledge architecture in Figure 9. In the final layer of our model, we employed a layer aggregation technique. The layer aggregation process utilizes the formula

$$h_v^{final} = (Z^{(0)}(v, :), h_v^{(1)}, h_v^{(2)}, h_v^{(3)})\Theta_{\text{cat}},$$

to effectively combine the information from the various layers. $n_{classes}$ denotes the number of output categories specific to the dataset. $\Theta_{\text{cat}} \in \mathbb{R}^{d_{eff} \times \text{n}_{\text{classes}}}, d_{eff} = d_1 + d_2 + d_3$.

## D.2 Evaluating the impact of increasing sketch-ratios.

To evaluate the impact of sketch ratio ($r = \frac{c_0}{n}$) on the accuracy of PGNN and Sketch-GNN under comparable conditions, we conduct additional experiments on the Cora and Citeseer datasets using the GCN architecture as the base. The results in Table 6 illustrate the performance of both methods across similar and increasing sketch ratios. We observe that the accuracies of our PGNN method is better than sketch-GNN for the Cora and the Citeseer dataset for similar sketch-ratios. The training loss comparison between PGCN and GCN methods across three citation datasets is shown in Figure 5. These results show that increasing the

Table 5: Comparison of node representations between the PGCN method and the Taylor series approximation of the GCN update rule.

| Dataset | Method | $e^{(1)}_{Taylor}$ **Layer 1** | $e^{(2)}_{Taylor}$ **Layer 2** |
|---------|--------|--------------|--------------|
| Cora | PGCN | 1.0105 | 0.9829 |
| Cora | Taylor | 1.0274 | 0.8807 |
| Citeseer | PGCN | 1.0075 | 0.6747 |
| Citeseer | Taylor | 1.1895 | 1.0110 |

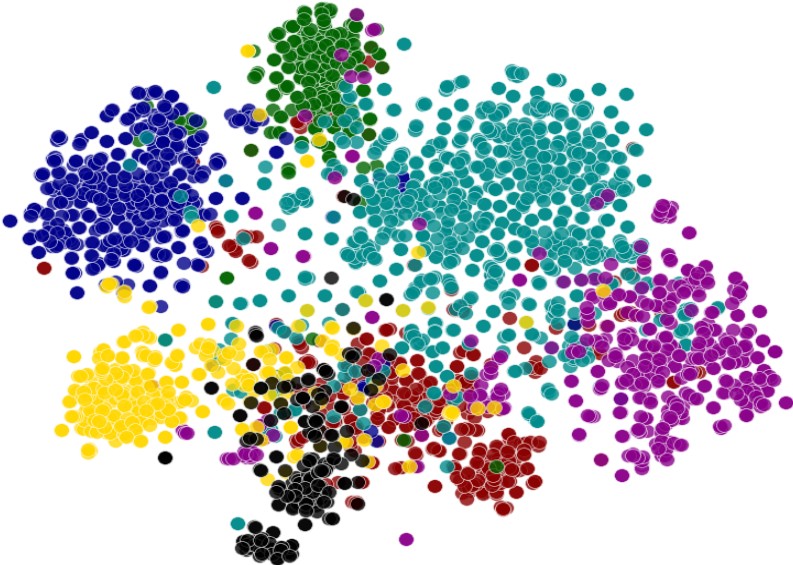

Figure 10: A t-SNE plot of the computed feature representations of the pre-trained PGCN at the first layer on the Cora dataset. Node colours denote classes.

sketch-ratio generally improves convergence speed, particularly for the Cora and Citeseer datasets. Notably, PGCN demonstrates significantly faster convergence compared to the baseline GCN. For Pubmed, PGCN exhibits convergence behavior comparable to GCN at sketch-ratios of 0.01 and 0.02.

Table 6: Accuracy Comparison of Sketch-GNN and PGNN on Cora and Citeseer

| Dataset | Sketch Ratio | Sketch-GNN Accuracy (% ± Std) | PGNN Accuracy (% ± Std) |
|---------|--------------|-------------------------------|-------------------------|
| Cora | 0.013 | 80.12 ± 1.04 | 80.65 ± 0.68 |
| Cora | 0.026 | 80.35 ± 0.71 | 80.52 ± 1.24 |
| Citeseer | 0.009 | 70.91 ± 0.93 | 70.23 ± 0.61 |
| Citeseer | 0.018 | 71.14 ± 0.59 | 71.21 ± 1.13 |

### D.3 Evaluating the Quality of Node Representations at each Layer.

In Table 5, we present a comparative analysis of node representations generated by the Taylor series approximation and the PGNN method with the GCN architecture. The results demonstrate that the PGNN method achieves lower error rates across individual layers for the Citeseer dataset. For the Cora dataset, PGNN exhibits reduced errors in the first layer, while the errors in the second layer are comparable to those

of the Taylor series approximation. The Taylor series approximation of the node representations and the node representations obtained from the PGCN method is utilized to compute $e_{Taylor}^{(l)} \left( \frac{\left\| X_{Taylor}^{(l)} - \hat{X}^{(l)} \right\|_F}{\left\| X_{Taylor}^{(l)} \right\|_F} \right)$. For the first layer, we have:

$$X^{(1)} = \sigma(CX^{(0)}\Theta^{(0)}).$$

The Taylor series approximations of the node representations at layer one and $i-$th column are given by $X_{Taylor}^{(l)}(:,i) = CX^{(0)}\Theta^{(0)}(:,i) + CX^{(0)}\Delta\Theta(:,i)$. The approximate node representations in the graph dimension $n$ by the PGCN method in the first layer is given by

$$\hat{X}^{(1)} = \text{Mean}\left\{ R^{(k)T} \left( (\tilde{\rho}^{(k)})^T \rho_1 Z^{(1)} \right) \right\} + [\bar{x}]_{n \times d_l}.$$

Mean refers to the element-wise mean over tensors.
We conducted additional experiments on Cora and Citeseer datasets to examine error accumulation for deeper GNNs. In Table 7, we present the relative error at layer $l$, defined as $e^{(l)} = \frac{\|X^{(l)} - \hat{X}^{(l)}\|_F}{\|X^{(l)}\|_F}$. Results

Table 7: Relative error ($e^{(l)}$) between GCN and PGCN representations across different network depths on Cora and Citeseer datasets, with and without jumping knowledge (JK) connections.

| Layers ($l$) | Cora | | Citeseer | |
|---|---|---|---|---|
| | without JK | with JK | without JK | with JK |
| 1 | 0.6744 | 0.6744 | 0.7835 | 0.7835 |
| 2 | 1.1222 | 1.1222 | 1.0893 | 1.0893 |
| 3 | 1.1270 | 1.1270 | 1.3990 | 1.3990 |
| 4 | 1.9854 | 1.9854 | 1.7829 | 1.7829 |
| 5 | 1.7450 | 1.7450 | 1.4365 | 1.4365 |
| 6 | 1.2477 | 1.0898 | 2.4158 | 1.0585 |

show that the relative error fluctuates across layers for both datasets. The error values remain identical with and without jumping knowledge connections for layers 1-5, as jumping knowledge only impacts the final layer aggregation. At layer 6, jumping knowledge connections significantly reduce the error (from 1.2477 to 1.0898 for Cora, and from 2.4158 to 1.0585 for Citeseer), effectively mitigating error accumulation in deeper architectures.

### D.4 Comparison of spectral properties of the sketches of the convolution matrices

We say that a matrix $B \in \mathbb{R}^{n \times n}$ is an $\epsilon$ approximation to matrix $A \in \mathbb{R}^{n \times n}$ if their quadratic forms have the form

$$\frac{x^T B x}{\epsilon} \leq x^T A x \leq \epsilon\, x^T B x \ \ \forall\, x \in \mathbb{R}^n.$$

The above equivalence implies the spectrum similarity between the two matrices (Courant-Fisher Theorem Chung (1997)). We present comparisons of the eigenvalues and eigenvectors of the convolution matrix $C$ and the equivalent convolution matrix $C_{eq} = PC$ for the Cora dataset in Figures 11a and 11b. The eigenvalues and eigenvectors of the matrix $C_{eq}$ closely align with those of $C$.

### D.5 Additional Experiments: Link Prediction

To further evaluate the effectiveness of our method beyond node classification, we conducted additional experiments on the task of link prediction using the **Cora**, **Citeseer**, **Pubmed**, and **ogbn-arxiv** datasets. The results below compare PGNN with the standard GCN baseline. We observe that PGNN outperforms the GCN method on all the four datasets.

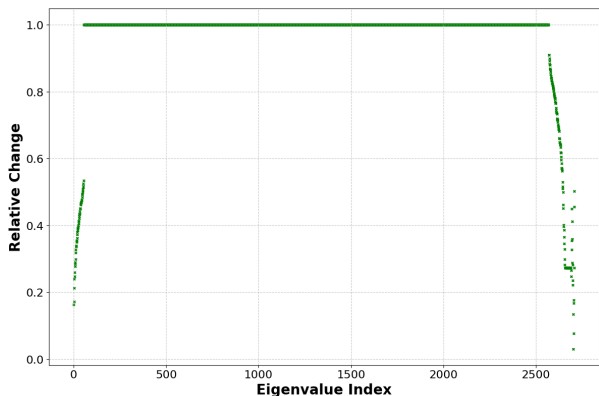
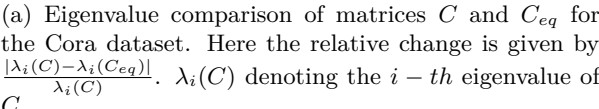
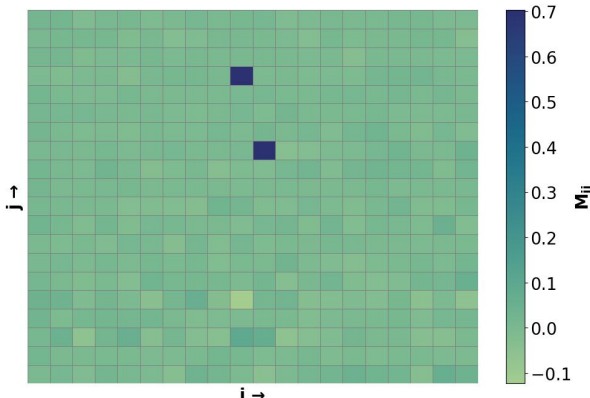

(a) Eigenvalue comparison of matrices $C$ and $C_{eq}$ for the Cora dataset. Here the relative change is given by $\frac{|\lambda_i(C) - \lambda_i(C_{eq})|}{\lambda_i(C)}$. $\lambda_i(C)$ denoting the $i-th$ eigenvalue of $C$.

(b) Comparison of Eigenvectors of $C$ and $C_{eq}$ for the Cora dataset, with the $ij$-th element $\mathrm{M}_{ij}$ quantified by $\phi_{ij}(C) - \tilde{\phi}_{ij}(C_{eq})$, where $\phi_{ij}(C)$ and $\tilde{\phi}_{ij}$ denote the $ij$-th component of the eigenvectors for $C$ and $C_{eq}$, respectively.

Figure 11: Comparison of eigenvalues and eigenvectors of $C$ and $C_{eq}$ for the Cora dataset.

Table 8: Link prediction performance comparison between PGNN and GCN across four citation datasets. Best Test AUC ($\pm$ Std) scores for each dataset are highlighted in **green**.

| Dataset | GCN | PGNN (Sketch Ratio) |
|---|---|---|
| ogbn-arxiv | $92.08 \pm 1.38$ | **94.74 $\pm$ 0.78** ($r = 0.10$) |
| Pubmed | $90.51 \pm 1.82$ | **92.66 $\pm$ 2.86** ($r = 0.04$) |
| Cora | $78.24 \pm 0.04$ | **81.90 $\pm$ 3.30** ($r = 0.03$) |
| Citeseer | $77.40 \pm 2.32$ | **86.48 $\pm$ 1.92** ($r = 0.03$) |

### D.6 Convergence Analysis: Loss and Accuracy in Node Classification

The convergence analysis for PGCN and PSAGE architectures on the Reddit dataset is presented in Figures 12 and 13, where we examine the evolution of training loss alongside training, validation, and test accuracy metrics throughout the optimization process.

## E Implementation Details

We outline the various implementation details with the hyper-parameter setups for experiments in section 5.

**Datasets.** Table 9 provides a comprehensive summary of the statistics for all datasets utilized in the experiments. The datasets ogbn-arxiv and ogbn-products were sourced from the Open Graph Benchmark (OGB)[1]. The Reddit dataset, a more streamlined variant of the original dataset by Hamilton and colleagues, was acquired through the PyTorch Geometric library[2]. For our research, we adhered to the conventional dataset divisions established by OGB and PyTorch Geometric.

**Code Frameworks.** The codes used for experimentation are made available at [3]. PGNN framework make use of the PyTorch library and the PyTorch Sparse library[4]. For the computation of the sketch of the input node feature matrix, the svd function from the Pytorch library is used. The Count-sketch technique implementation is taken from the repository[5]. All of the above code repositories we used are licensed under

---

[1] https://ogb.stanford.edu/

[2] https://github.com/pyg-team/pytorch_geometric

[3] https://github.com/ast-fri/PGNN

[4] https://github.com/rusty1s/pytorch_sparse

[5] https://github.com/johnding1996/Sketch-GNN-Sublinear

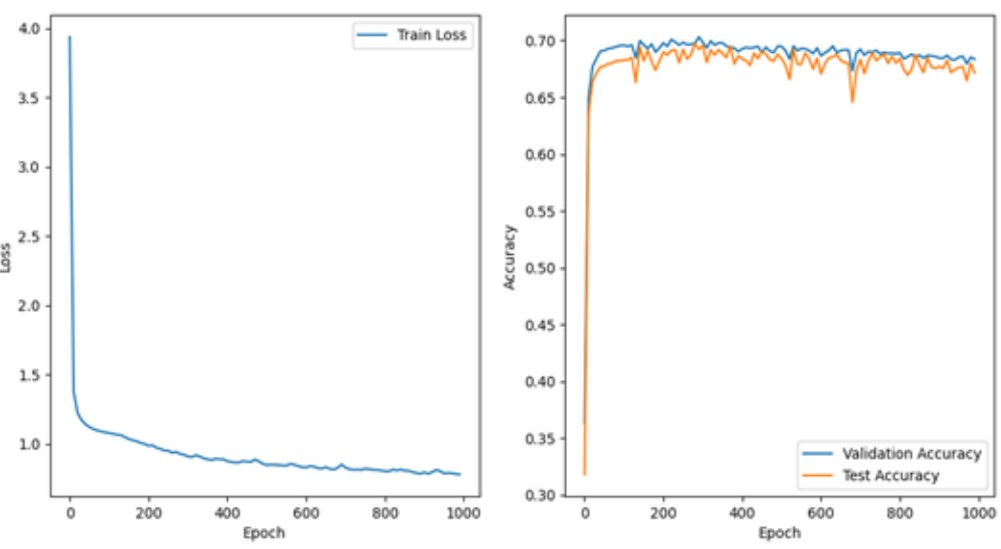

Figure 12: Training convergence behavior of PGCN method for node classification downstream task for Reddit.

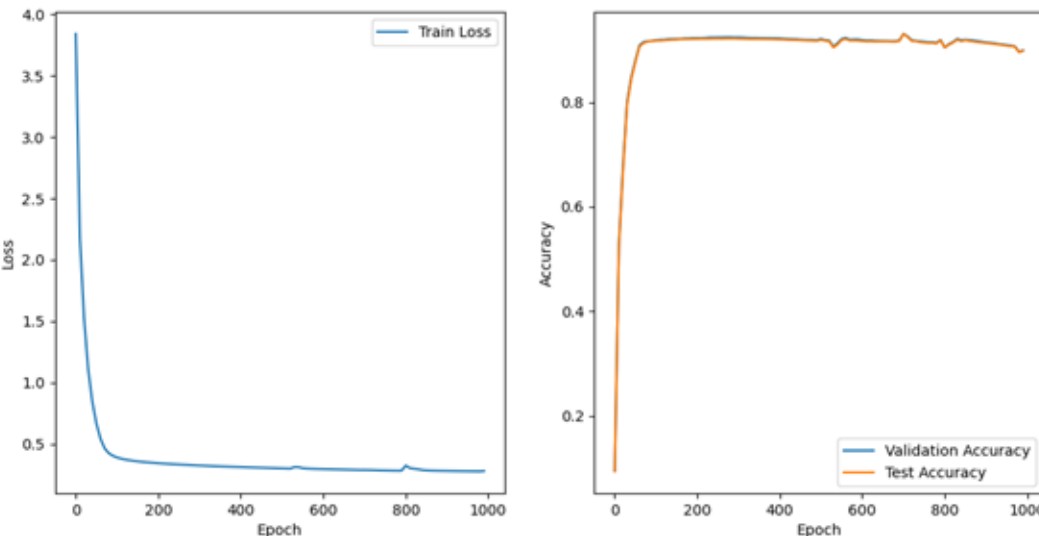

Figure 13: Training convergence behavior of PSAGE method for node classification downstream task for Reddit.

the MIT license.

## E.1 Hyperparameters

We conducted a comprehensive hyperparameter search across all model architectures. Learning rates were tuned in the range $[0.001, 0.05]$, and weight decay values in the range $[10^{-5}, 4 \times 10^{-2}]$. Network

| Dataset | Cora | Citeseer | Pubmed | ogbn-arxiv | Reddit | ogbn-products | ogbn-papers100M |
|---|---|---|---|---|---|---|---|
| Task | node | node | node | node | node | node | node |
| Setting | transductive | transductive | transductive | transductive | transductive | transductive | transductive |
| Label | single | single | single | single | single | single | single |
| Metric | accuracy | accuracy | accuracy | accuracy | accuracy | accuracy | accuracy |
| # of Nodes | 2,708 | 3,327 | 19,717 | 169,343 | 232,965 | 2,449,029 | 111,059,956 |
| # of Edges | 5,429 | 4,732 | 44,338 | 1,166,243 | 11,606,919 | 61,859,140 | 1,615,685,872 |
| # of Features | 1,433 | 3,703 | 500 | 128 | 602 | 100 | 128 |
| # of Classes | 7 | 6 | 3 | 40 | 41 | 47 | 172 |

Table 9: Detailed Overview of the graph datasets utilized in experiments.

architectures employed hidden dimensions of $\{128, 150, 192, 256\}$ with either two or three layers. Dropout rates were varied across $\{0.0, 0.2, 0.5\}$, and normalization strategies included no normalization, BatchNorm, or LayerNorm. For PGNN-specific parameters, both the order $r$ and number of sketches $k$ were set from $\{2, 3\}$. The regularization parameters $\alpha$, $\alpha_0$, and $\beta_0$ were tuned over $\{0, 0.5, 1\}$, while $\beta$, $\beta_2$ were constrained to at most $\lceil 0.1n \rceil$.

| Dataset | Model | Learning Rate | Weight Decay | Hidden Dim | Num Layers | Dropout | Norm Type |
|---|---|---|---|---|---|---|---|
| ogbn-arxiv | PGCN | 0.01 | 1e-5 | 128 | 2 | 0.0 | BatchNorm |
| | PSAGE | 0.01 | 1e-5 | 128 | 2 | 0.2 | BatchNorm |
| | PGAT | 1e-3 | 1e-4 | 128 | 2 | 0.2 | BatchNorm |
| | PSGC | 0.02 | 4e-5 | 128 | 2 | 0.0 | None |
| Reddit | PGCN | 0.008 | 2e-5 | 128 | 2 | 0.0 | None |
| | PSAGE | 0.008 | 2e-5 | 128 | 2 | 0.0 | None |
| | PGAT | 0.001 | 1e-5 | 128 | 2 | 0.0 | LayerNorm |
| | PSGC | 0.008 | 2e-5 | 192 | 2 | 0.0 | None |
| Cora | PGCN | 0.01 | 2e-3 | 128 | 3 | 0.0 | None |
| | PSAGE | 0.05 | 1e-4 | 256 | 2 | 0.5 | None |
| | PGAT | 0.002 | 1e-3 | 128 | 3 | 0.0 | None |
| | PSGC | 0.01 | 1e-3 | 128 | 2 | 0.0 | None |
| Citeseer | PGCN | 0.01 | 9e-4 | 128 | 2 | 0.0 | None |
| | PSAGE | 0.005 | 5e-4 | 256 | 2 | 0.2 | None |
| | PGAT | 0.001 | 1e-4 | 128 | 2 | 0.0 | None |
| | PSGC | 0.01 | 5e-4 | 128 | 2 | 0.0 | None |
| Pubmed | PGCN | 0.004 | 6e-3 | 150 | 2 | 0.2 | None |
| | PSAGE | 0.03 | 2e-4 | 150 | 2 | 0.5 | None |
| | PGAT | 0.004 | 6e-3 | 150 | 2 | 0.2 | None |
| | PSGC | 0.005 | 4e-2 | 128 | 2 | 0.2 | None |

Table 10: Hyperparameters for Different Models and Datasets

**Computational Infrastructures.** The experiments demonstrate the scalability of our model across different computational setups. Small-scale datasets, including Cora, Pubmed, and Citeseer, were processed on an Nvidia A30 GPU with 24 GB memory, while larger datasets like ogbn-arxiv and Reddit were trained and evaluated on an Nvidia A100 GPU with 80GB memory for faster training and inference. The system was equipped with an Intel Xeon Platinum CPUs and 512GB of RAM, ensuring efficient execution of large-scale graph neural network experiments..

**Setup of PGNN:** In our experimental setup, we have designated at most 1000 epochs for each run, with 10 runs to ensure statistical significance, more details about hyperparameters for different experiments are listed in Table 10. We keep the order $q$ and the number of sketches $k$ equal. The sketch-ratio

of 0.018 used for the citeseer is the same as mentioned from the paper Ding et al. (2022), sketch-ratio of 0.02 against 0.026 is used for the cora dataset to demonstrate the effectiveness of the proposed method. For the Pubmed dataset, we selected a lower sketch ratio of 0.01. This choice aligns with the general principle that as graph size increases, the sketch ratio or the effective number of components for preserving variance decreases. Extensive experimentation confirmed that a sketch ratio of 0.01 was sufficient to achieve good classification accuracy. For PGAT, we employ 2 attention masks. The training times of PGNN were not compared with the existing sketch-based method, Sketch-GNN (Ding et al., 2022), due to observed discrepancies in node classification accuracy from the implementation available in the repository 5.

