# OpenReview forum: "Proper Orthogonal Decomposition for Scalable Training of Graph Neural Networks"
_TMLR — Accepted by TMLR_

### Review · Reviewer_xgkJ · 2025-08-16

**Summary Of Contributions:**

**Summary:**
The paper introduces a sketch-based method using Proper Orthogonal Decomposition (POD) to construct low-dimensional approximations of graph data and convolution operations. These approximations reduce both memory usage and runtime, making them particularly effective for large-scale graph processing.

Key contributions include:
- A POD-based method for low-dimensional graph approximations.
- Application to linear GNNs such as SGC.
- Extension to handle non-linearities via Tensor sketches.
- Solutions to challenges such as:
  - Restrictions on the column space of learnable functions.
  - \(O(n)\) computation cost requirements.
- Theoretical guarantees:
  - Optimality of POD for linear updates.
  - Error bounds for the proposed method.
- Experimental validation:
  - Strong performance on large-scale transductive graph learning tasks (e.g., OGBN datasets).
  - Results on large graph benchmarks and specially on OGBN-Papers100M, with sketch ratios as low as \(10^{-6}\).

**Strengths**
1. Addresses an important problem: scaling GNNs to real-world, massive graphs.
2. Effectively reduces both memory and runtime costs with minimal accuracy loss.
3. Method is well-crafted, considering multiple aspects of the scalability challenge, and grounded in strong theory.
4. Provides clear theoretical guarantees.
5. Experimental results demonstrate strong performance across multiple datasets.
6. Improves over prior sketch-based GNNs, achieving significantly lower sketch ratios (down to \(10^{-6}\) on OGBN-Papers100M).

**Audience:**

Yes

**Audience Explanation:**

Learning on graphs has become a major subfield within Machine Learning, and the challenge of scaling methods to very large graphs is both well-motivated and of significant practical importance. Addressing this problem has direct implications for real-world applications, where datasets often contain millions or even billions of nodes and edges.

The authors’ approach falls within the family of sketch-based methods, which represent a promising direction for improving scalability. Sketching techniques provide a principled way to reduce memory and runtime complexity while preserving key structural information, and this paper leverages that idea in a thoughtful manner. By applying Proper Orthogonal Decomposition (POD) and extending it with Tensor sketches and Local Sensitive Hashing, the work advances the current state of sketch-based graph learning and demonstrates its potential to handle graphs at the scale of OGBN-Papers100M.

Overall, I believe this paper will be of interest to the audience of TMLR. The combination of theoretical grounding, algorithmic novelty, and empirical validation makes it a valuable contribution to the community, particularly for those interested in scalable graph representation learning.

**Broader Impact Concerns:**

The paper does not explicitly discuss broader impact. In my view, this work is primarily methodological and focuses on improving the scalability of GNNs through sketch-based techniques. As such, its broader societal impact may be limited. Overall, I do not see significant ethical or societal concerns specific to this work.

**Claims And Evidence:**

Yes

**Claims Explanation:**

Most of the claims in the paper are well-supported. Below is a breakdown of positive aspects and areas for improvement:

**Positive Points**
1. The addressed problem is theoretically well-motivated, and the proposed method provides a convincing approach to the challenges of large-scale graph processing.
2. The method is mostly well-explained, with the mathematical formulation making sense overall.
3. Theoretical guarantees are adequately addressed, particularly through Theorems 1 and 2.
4. Experimental results are generally convincing and demonstrate:
   - Minimal performance drop despite substantial improvements in memory and runtime efficiency.
   - A strong selection of datasets, with appropriate scale to validate the claims.

**Areas for Improvement / Clarifications Needed**
1. **Clarity of Explanation:** Section 4.1 was difficult to follow. While the mathematics is correct, it would benefit from additional intermediate steps in the derivations. If referencing external resources, please indicate explicitly which theorem or result leads to the final equation.
2. **Experimental Results:** While overall convincing, some issues stand out:
   - In Table 1 (GAT model on OGBN-Products), why does the lower sketch ratio result in an OOM error?
   - In Table 3, SGC and MLP appear as OOM, but results are reported for GCN. For the MLP specifically, batch size can be adjusted to process small sizes of nodes at a time and it should scale to very small memory sizes. Have these points considered in the analysis?
   - Some improvements seem marginal, e.g., a reduction from 0.026 to 0.02 compared to Sketch-GNN, which may not be statistically significant.

**Requested Changes:**

1. Please revise Section 4.1 to provide more detailed explanations and intermediate steps, making the mathematical derivations easier to follow.
2. Please expand on the description of the experimental setups, and clarify or adjust the experiments involving the MLP baseline to ensure consistency and completeness.

---

### Review · Reviewer_F3wH · 2025-08-17

**Summary Of Contributions:**

The paper introduces PGNN, a sketch-based training framework for GNNs built on Proper Orthogonal Decomposition (POD). The key idea is to precompute a low-dimensional subspace (via POD on the augmented feature matrix) and train entirely within this subspace, avoiding the online sketch updates used by prior sketch-based methods. The authors (i) derive PGNN update rules for several architectures (SGC/GCN/SAGE/GAT), (ii) prove POD-optimality for the linearized (SGC) case and provide a per-layer error bound, and (iii) show empirical scalability with materially lower sketch ratios than Sketch-GNN while maintaining competitive accuracy and reduced memory/time. Experiments span Cora/Citeseer/PubMed, OGBN-Arxiv/Products/ Papers100M, and Reddit, with notable results such as r=0.05 on Reddit vs r=0.3 for Sketch-GNN, and a ~339× per-epoch speedup on Papers100M at small sketch ratios.

Pros:
1. Clear algorithmic framing and practical recipe (preprocess → train with fixed subspace → LSH-based selective loss evaluation).
2. Strong theoretical support: POD optimality for SGC and error propagation bound.
3. low sketch ratios (e.g., 0.003 on OGBN-Products) with competitive accuracy; substantial memory/time savings on large graphs.
4. Broad architectural coverage and ablations across datasets; link prediction results included.

Cons
1. Theory mainly covers linearized GNNs; guarantees for nonlinear message passing are indirect.
2. LSH/triplet-loss machinery introduces extra hyperparameters and complexity; sensitivity is only briefly discussed.

**Audience:**

Yes

**Audience Explanation:**

Scalable GNN training is a very important topic for Graph foundation model.

**Broader Impact Concerns:**

None.

**Claims And Evidence:**

Yes

**Claims Explanation:**

The paper’s central claims that POD-based subspace training enabling lower sketch ratios and improved efficiency with comparable accuracy are supported by: (1) a clean derivation for the linearized case and an explicit layer-wise error bound; (2) consistent empirical results across small/medium OGB datasets and large-scale Reddit/Products/Papers100M showing accuracy close to or better than Sketch-GNN at significantly smaller r, lower memory, and faster epochs; and (3) concrete runtime/memory tables and figures. Limitations are acknowledged (e.g., accuracy drop on OGBN-Arxiv and the lack of convolution sketch for GAT). While nonlinear-theory coverage is limited, the empirical evidence is methodical and convincing for the stated scope.

**Requested Changes:**

1. Provide a brief theoretical discussion clarifying why POD optimality does not extend to nonlinear cases, and add an empirical stress test (deeper GCN/GAT) to illustrate error accumulation and the effect of Jumping Knowledge within PGNN.
2. For key tables (Reddit/Products/Arxiv), clarify hyperparameter tuning and compute budgets for all baselines.

---

### Review · Reviewer_8W9r · 2025-09-22

**Summary Of Contributions:**

The paper introduces PGNN, a novel sketch-based framework designed to scale Graph Neural Networks to very large graphs by leveraging Proper Orthogonal Decomposition (POD) and Tensor Sketching. PGNN approximates standard GNN message passing by projecting node representations onto a low-dimensional subspace, significantly reducing memory and computational cost. It integrates count-sketch and Tensor Sketch techniques to efficiently compress both node features and convolution matrices, while a learnable projection matrix enhances flexibility beyond the fixed POD subspace. Additionally, PGNN employs a locality-sensitive hashing approach to evaluate node classification losses sublinearly, focusing on nodes with poor predictions. Extensive experiments on benchmark datasets, including massive graphs with over 100 million nodes, demonstrate that PGNN achieves competitive or superior accuracy compared to state-of-the-art methods, while dramatically improving memory efficiency and training speed. The framework is supported by theoretical guarantees for projection optimality and error bounds, providing strong evidence for its effectiveness.

Strengths:
- Elegant integration of POD and sketching techniques for scalable GNNs.
- Demonstrated efficiency and accuracy on datasets up to 111 million nodes.
- Strong theoretical grounding with error bounds (Theorem 2) and optimality proof for POD projection (Theorem 1).

Weaknesses:
- Limited evaluation on tasks beyond node classification (only brief link prediction experiments).
- Complexity of the method (multiple sketches, FFT operations, LSH integration) may hinder adoption for practitioners unfamiliar with these techniques.

**Audience:**

Yes

**Audience Explanation:**

Scaling GNNs to large graphs is a highly relevant problem in machine learning and graph analytics. Researchers and practitioners working with large graph datasets (e.g., social networks, citation networks, e-commerce recommendation systems) would benefit from PGNN’s contributions in reducing memory and runtime while preserving accuracy.

**Broader Impact Concerns:**

No ethical concerns.

**Claims And Evidence:**

Yes

**Claims Explanation:**

The paper provides comprehensive experimental results across multiple benchmark datasets, including both small citation graphs and large-scale graphs. Comparisons with state-of-the-art methods (GCond, Graph Coarsening, Sketch-GNN, GraphSAINT, VQ-GNN) support the claims regarding accuracy, memory efficiency, and training time. Theoretical guarantees are provided for approximation errors and POD optimality, and empirical studies on error propagation and t-SNE visualization support the method’s robustness.

**Requested Changes:**

- Clarify preprocessing complexity: Provide a more detailed analysis of preprocessing time with respect to graph size (currently described but not fully quantified for intermediate scales).
- Broaden evaluation: Consider including additional tasks (e.g., graph classification) or more detailed link prediction experiments to show generality.
- Simplify exposition: Some sections (e.g., Tensor Sketch-based updates, LSH-based loss evaluation) are mathematically dense and could benefit from clearer illustrations or pseudocode to improve accessibility.
These changes are not critical to validate the main claims but would strengthen clarity, generality, and reproducibility.

---

### Decision · Action_Editor_kzUL · 2025-12-03

**Recommendation:** Accept as is

**Audience:**

Yes

**Audience Explanation:**

Research on scalable algorithms applicable to massive graphs is a central topic in the field of Graph Neural Networks (GNNs). Therefore, the findings presented in this paper are expected to be of interest to the TMLR audience, especially those researching GNNs.

All three reviewers agreed on this point, concurring that scalable GNN methods represent an important area in machine learning and graph analysis with implications for real-world applications.

**Claims And Evidence:**

Yes

**Claims Explanation:**

This paper proposes PGNN, a novel sketch-based GNN model for node classification tasks. The proposed algorithm combines Count Sketch and Tensor Sketch techniques. In addition, the proposed method improves computational efficiency by optimizing the loss function calculation using Locality Sensitive Hashing.

The submission makes the following primary claims:
- PGNN achieves a form of error optimality for the linear case.
- The approximation error of node representations introduced by the PGNN update rule is evaluated.
- PGNN improves computational efficiency while achieving accuracy comparable to existing methods.

Three reviewers assessed the submission and agreed that these claims are supported by evidence. While the reviewer team had some discussion regarding the sufficiency of the theoretical analysis, the authors addressed these concerns in the revised version.